# BRCA1 binds TERRA RNA and suppresses R-Loop-based telomeric DNA damage

Jekaterina Vohhodina [1,2✉], Liana J. Goehring[1], Ben Liu[1,2], Qing Kong[1,2], Vladimir V. Botchkarev Jr.[1,2], Mai Huynh[1], Zhiqi Liu[1], Fieda O. Abderazzaq[1,2], Allison P. Clark[1,2], Scott B. Ficarro[1,3,4,5], Jarrod A. Marto [1,3,4,5], Elodie Hatchi [1,2] & David M. Livingston [1,2✉]

R-loop structures act as modulators of physiological processes such as transcription termination, gene regulation, and DNA repair. However, they can cause transcription-replication conflicts and give rise to genomic instability, particularly at telomeres, which are prone to forming DNA secondary structures. Here, we demonstrate that BRCA1 binds TERRA RNA, directly and physically via its N-terminal nuclear localization sequence, as well as telomere-specific shelterin proteins in an R-loop-, and a cell cycle-dependent manner. R-loop-driven BRCA1 binding to CpG-rich TERRA promoters represses TERRA transcription, prevents TERRA R-loop-associated damage, and promotes its repair, likely in association with SETX and XRN2. BRCA1 depletion upregulates TERRA expression, leading to overly abundant TERRA R-loops, telomeric replication stress, and signs of telomeric aberrancy. Moreover, BRCA1 mutations within the TERRA-binding region lead to an excess of TERRA-associated R-loops and telomeric abnormalities. Thus, normal BRCA1/TERRA binding suppresses telomere-centered genome instability.

[1] Department of Cancer Biology, Dana-Farber Cancer Institute, Boston, MA, USA. [2] Department of Genetics, Harvard Medical School, Boston, MA, USA. [3] Blais Proteomics Center, Dana-Farber Cancer Institute, Boston, MA, USA. [4] Department of Oncologic Pathology, Dana-Farber Cancer Institute, Boston, MA, USA. [5] Department of Pathology, Brigham and Women's Hospital, Harvard Medical School, Boston, MA, USA. ✉email: jekaterina_vohhodina@dfci.harvard.edu; david_livingston@dfci.harvard.edu

Telomeres are nucleoprotein structures that prevent the ends of chromosomes from becoming fused, degraded, and recognized as sites of DNA damage. They contain arrays of long, double-stranded DNA hexamer repeats (5′-TTAGGG-3′ in vertebrates), telomere-specific, six-subunit protein complexes, i.e., "shelterins", and telomerase complexes[1]. Another telomere maintenance element is the long noncoding RNA, TERRA, i.e., a telomeric repeat-containing RNA.

TERRA is a product of RNA polymerase II (RNAPII)-transcribed telomeric DNA that participates in telomere length regulation and chromosome end protection[2–4]. G-rich TERRA molecules are transcribed from subtelomeric regions toward the ends of chromosomes, using a C-rich telomeric strand as template[2,3]. TERRA transcription is driven from repetitive CpG-rich promoter sequences located in the subtelomeric regions of at least half of human chromosomes[5]. CpG-rich TERRA promoters are regulated by the methylation driven by the DNA methyltransferases, DNMT1 and DNMT3b, depletion of which leads to upregulated TERRA transcription[5].

Telomeres are hotspots for the formation of secondary genomic structures, such as T-loops, D-loops, G-quadruplexes (G4), and R-loops[6,7]. Indeed, G-rich TERRA RNA, itself, forms R-loops with the C-rich telomeric strand[8]. In fact, formation of TERRA R-loops in cancer cells that maintain their telomeric lengths by the alternative lengthening of telomeres (ALT) pathway, is a beneficial process, since TERRA R-loops support homologous recombination (HR)-based telomeric replication[9].

That said, an excess of TERRA R-loops also leads to replication stress and genomic instability[9]. TERRA R-loop overproduction occurs in the cells of patients with the immunodeficiency, centromeric instability, and facial abnormalities (ICF) syndrome. These patients carry mutations in the DNMT3b gene, and their cells manifest accelerated telomeric shortening and premature senescence[10].

TERRA abundance in yeast is, in part, controlled by the Rat1p 5′–3′ exonuclease, which induces its degradation[11]. Depletion of the Rat1p-orthologue, XRN2, in gorilla cells also leads to TERRA accumulation[12]. A role for human XRN2 in TERRA regulation has not yet been reported.

A telomeric role for BRCA1 has been studied. It associates with the shelterin proteins, TRF1 and TRF2, and colocalizes with telomeric DNA in ALT cells[13,14]. It also associates with the BLM helicase, which maintains efficient telomeric replication in ALT cells[14,15].

Other BRCA1 telomeric functions involve the regulation of (a) telomerase activity, (b) length of the G-rich telomeric 3′ overhang, and (c) processing of dysfunctional telomeres together with CtIP[13,16,17]. Thus, BRCA1 mutation-bearing cells exhibit increased telomeric perturbations, resulting in accumulated telomere DNA damage and premature senescence[18,19].

In addition to BRCA1 functions in DNA damage repair, it also participates in eliminating DNA damage-promoting R-loops, together with FUS and/or senataxin (SETX)[20–22]. BRCA1 also resolves R-loops that form at transcription start/termination sites during negative elongation factor-mediated RNAPII pausing[23]. Similarly, BRCA1-modulated R-loop mitigation upstream of the estrogen receptor α locus can influence luminal gene transcription and epithelial differentiation, conceivably repressing BRCA1-associated tumorigenesis[24].

In light of these findings, we asked whether BRCA1 contributes to the maintenance of telomeres through TERRA-based R-loop regulation. Here, we show that BRCA1 physically interacts with TERRA RNA in a cell cycle-dependent manner upon the formation of R-loops at telomeres. We further find that R-loop-dependent binding of BRCA1 to CpG-rich TERRA promoters suppresses TERRA transcription and TERRA R-loop formation, thus suppressing telomere-associated DNA damage development. We have also identified a specific domain of the BRCA1 protein that is responsible for its direct interaction with TERRA RNA, and observed that mutations in this region interfere with TERRA binding and trigger increased TERRA-based R-loop formation, resulting in telomeric replication stress and telomeric dysfunction.

Thus, we report new and essential roles for BRCA1 in the regulation of TERRA, thereby safeguarding telomere integrity and, perhaps, contributing to the suppression of cancer development.

## Results

**BRCA1 association with telomeres and XRN2 is R-loop dependent.** Reciprocal co-immunoprecipitation (co-IP) experiments revealed an interaction of BRCA1 with all members of the shelterin complex (Fig. 1a and Supplementary Fig. 1a). BRCA1 also localized to telomeres in a cell cycle-dependent manner, increasing in G1/S and becoming maximal in late S–G2/M phases, as reflected by BRCA1 chromatin immunoprecipitation (ChIP) and BRCA1 immunofluorescence-telomere DNA FISH analysis in synchronized T98G cells (Fig. 1b and Supplementary Fig. 1b–d).

To further investigate telomere-related BRCA1 functions, we performed BRCA1 and TRF2 IPs and searched for BRCA1 and/or TRF2-associated proteins by nanoLC–MS (Supplementary Fig. 1e). The analysis revealed 244 BRCA1- and 201 TRF2-interacting proteins, of which 90 were mutual (Fig. 1c and Supplementary Table 1). Functional annotation of these particular proteins revealed that the vast majority were DNA- or RNA-associated proteins, a number of which were 5′-, and 3′-UTR RNA processing factors (Supplementary Fig. 1f). For example, we detected FUS, which binds G-quadruplex structures at telomeres and increases RNAPII pausing[25]. FUS also colocalizes with BRCA1 at sites of transcription-associated DNA damage, including at R-loops[21].

Both BRCA1 and TRF2 also associate with members of the heterogeneous nuclear ribonucleoprotein family (hnRNPs), including hnRNPUL1, hnRNPAB, and hnRNPD, known to regulate chromatin remodeling, transcription, and DNA repair[26–28]. Thus, the results of our IP analyses suggest that RNA processing proteins and BRCA1 cooperate at telomeres.

One dually BRCA1/TRF2-bound protein that we selected for further investigation was XRN2, a 5′–3′ exoribonuclease implicated in RNAPII transcription termination[29]. XRN2 is also known to cooperate with SETX to resolve R-loops at G-rich transcription pause sites[30].

Our reciprocal co-IPs further confirmed that XRN2 interacts with BRCA1 and TRF2, as well as with other members of the shelterin complex (Fig. 1d and Supplementary Fig. 1g). The 1005–1313 aa BRCA1 region is likely responsible for the BRCA1–XRN2 association, since it was the only GST-tagged BRCA1 segment[31] that bound to XRN2 (Supplementary Fig. 1h, i).

Since both XRN2 and BRCA1 participate in the resolution/stabilization of R-loops[22,30], and depletion of these genes and of TRF2 resulted in R-loop accumulation (Supplementary Fig. 1j, k), we asked whether the interaction of BRCA1 and XRN2 with the main telomere DNA-associated proteins, TRF2, TRF1, and POT1, depends upon R-loop formation. BRCA1/SETX R-loop-dependent binding[22] served as a positive control (Fig. 1e).

Thus, we performed a series of co-IPs from cells that did or did not ectopically express RNase H1[30] (Supplementary Fig. 1l). It appeared that interactions of the shelterin proteins with BRCA1 and XRN2 were predominantly R-loop dependent (Fig. 1e).

Furthermore, in addition to telomeric repeats, we detected R-loop-dependent BRCA1 binding to the subtelomeric DNA

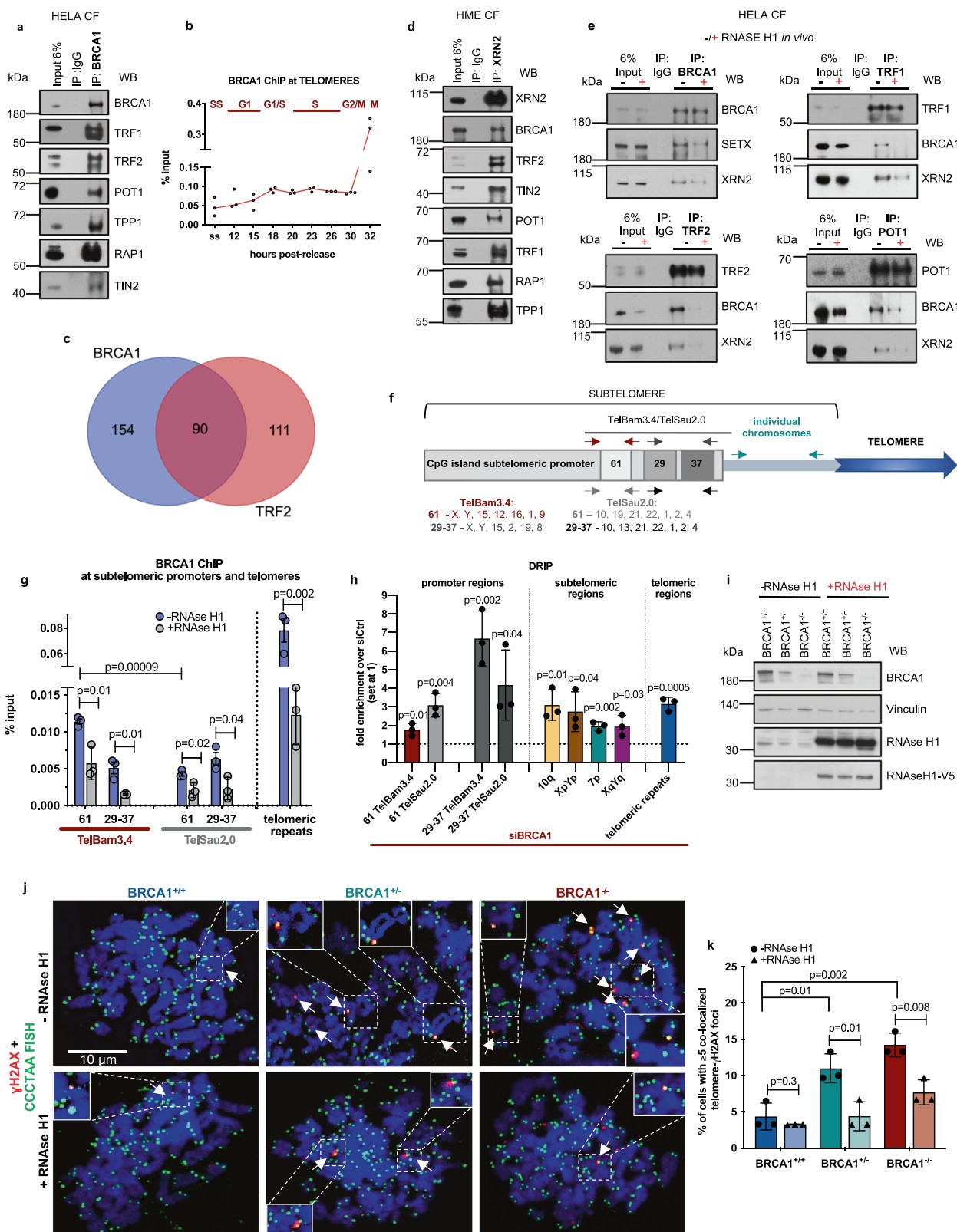

sequences, TelBam3.4 and TelSau2.0, which are located at multiple chromosome ends and represent TERRA CpG-island subtelomeric promoters[5] (Fig. 1f, g and Supplementary Fig. 1m). BRCA1 ChIP revealed predominantly R-loop-associated BRCA1 recruitment to conserved "61-29-37" repeats of these sequences, suggesting that BRCA1 binding to subtelomeric promoters and telomeric repeats is heavily driven by R-loop formation.

Consequently, BRCA1 depletion led to an increased presence of R-loops across distal and proximal subtelomeric regions and telomeric tracts (Fig. 1h).

**BRCA1 is required for telomere integrity.** In light of these findings, we asked whether BRCA1 functions in the repair of

**Fig. 1 BRCA1 interaction with the shelterin proteins and XRN2 at telomeres is R-loop dependent. a** Interaction of BRCA1 with the shelterin complex in HeLa cells. The chromatin fraction (CF) was IPd with BRCA1 antibody, and the immune complex was analyzed by immunoblotting. $n = 5$ independent experiments. **b** BRCA1 ChIP analyses of telomeric regions performed in synchronized T98G cells. Each dot represents a biological replicate ($n = 3$), and the median values are connected by a line, ss - serum-starved. **c** A Venn diagram revealing an overlap between proteins ($n = 90$) found in the endogenous pulldowns (CF) of BRCA1 and TRF2. **d** A pulldown revealing interaction of endogenous XRN2 with BRCA1 and the shelterin complex, using XRN2 antibody from CFs of HME cells. $n = 4$ independent experiments. **e** R-loop-dependent interactions between BRCA1, XRN2, and the shelterin proteins. CFs from ±RNAse H1-treated HeLa cells were used for IPs of endogenous BRCA1, TRF2, TRF1, and POT1. $n = 5$ independent experiments. **f** Schematic of CpG-island promoters and telomeric regions. Arrows indicate positions of primers used for ChIP, DRIP, and qRT-PCR. **g** BRCA1 ChIP analyses of promoter and telomeric regions performed in ±RNAse H1-treated U2OS cells. Bars represent the average value from $n = 3$ biological replicates for each sample, ±SD. $P$ values were obtained using a two-tailed Student's $t$ test. **h** DRIP analyses of promoter-telomeric regions performed in siCtrl and BRCA1-depleted U2OS cells. Bars represent the average value from $n = 3$ biological replicates for BRCA1-depleted sample over siCtrl, set at 1, ±SD. $P$ values were calculated using a two-tailed Student's $t$ test. **i** Immunoblot of CRISPR-modified HME cells confirming efficient transfection with V5-tagged RNAse H1-encoding vector. $n = 3$ independent experiments. **j** Representative images of cytospun HME CRISPR metaphase spreads in **i**, which were stained with an antibody for γH2AX and for telomeric DNA. Cells were mock- or RNAse H1-treated prior to the assay. Chromosome ends displaying γH2AX signals are identified by arrows. $n = 3$ independent experiments. **k** Mean percentage of CRISPR HME BRCA1$^{+/+}$, BRCA1$^{+/-}$, and BRCA1$^{-/-}$ cells with ≥5 positive γH2AX-telomere DNA colocalizations and the effects of RNAse H1 treatment from three independent experiments. At least $n = 30$ chromosome spreads scored/experiment, ±SD. $P$ values were obtained using a two-tailed Student's $t$ test. Source data are provided as a Source data file.

telomeric DNA damage, perhaps partially caused by unresolved R-loops. Specifically, we evaluated the state of telomeric integrity in the presence and absence of overexpressed RNAse H1 in our CRISPR-modified, isogenic monoallelic ($^{+/-}$) and biallelic ($^{-/-}$) BRCA1 knockout human mammary epithelial (HME) cells (Fig. 1i–k and Supplementary Fig. 2a–c). BRCA1$^{+/-}$ and BRCA1$^{-/-}$ cells demonstrated higher levels of telomeric γH2AX in metaphase spreads by comparison with BRCA1$^{+/+}$ cells, which were significantly reduced following RNAse H1 exposure, implying that BRCA1 is involved in preventing or repairing R-loop-driven damage at telomeres (Fig. 1i–k).

We also observed elevated levels of telomeric γH2AX in HME cells transiently depleted of BRCA1 via a doxycycline-inducible hairpin[32], as well as in XRN2-deficient cells (Supplementary Fig. 2d–h). Taken together, these data suggest that functional BRCA1 and XRN2 are essential for maintenance of telomeric integrity[13,14].

**BRCA1 interacts with TERRA in an R-loop-dependent manner.** Since mammalian G-rich TERRA is only transcribed from the C-rich telomeric strand[2,3] (Fig. 2a) and predictably forms RNA–DNA hybrids with it, we tested the affinity of BRCA1, XRN2, and SETX for C-rich and G-rich telomeric DNA sequence-imitating biotinylated probes (Fig. 2b). The G-rich strand binding protein, POT1, served as a positive control[1] (Fig. 2b). All three proteins exhibited strong interactions with the C-rich strand probe compared to either the G-rich or an unspecific oligonucleotide (Fig. 2b), suggesting the preferential binding of these proteins to the TERRA hybrid-forming DNA strand.

We next asked whether BRCA1, SETX, XRN2, and other DNA damage-associated proteins interact with TERRA RNA itself. Thus, an RNA pulldown was performed using two RNA probes, the TERRA-mimicking sequence (UUAGGG)$_8$ or its antisense equivalent (CCCUAA)$_8$ (Fig. 2c). Among the known TERRA-interacting proteins, TRF1 and TRF2 were enriched in a UUAGGG pulldown, but not RPA34, confirming previous studies[33] (Fig. 2c). BRCA1, SETX, XRN2, and other DNA damage response proteins, 53BP1 and MDC1, were also identified amidst the TERRA-bound proteins (Fig. 2c). Moreover, in keeping with the aforementioned results, we observed strong binding of BRCA1, SETX, and XRN2 to TERRA hybrids, composed of the TERRA probe and the C-rich DNA strand (Fig. 2d and Supplementary Fig. 3a), suggesting that an association of these proteins with telomeres is linked to TERRA

R-loop formation (Figs. 2d and 1e). XRN2 signal in RNAse H1-treated sample is likely the result of a nonspecific binding.

To further test whether BRCA1 interacts with endogenous TERRA in an R-loop-dependent manner, we performed an RNA immunoprecipitation (RIP) using anti-BRCA1 antibody followed by slot blot analysis, using a TERRA-specific probe in mock- or RNAse H1-treated U2OS cells (Fig. 2e and Supplementary Fig. 3b). Since exposure to RNAse H1 decreases TERRA abundance[9], we adjusted the amount of loaded RNA in the slot blot analyses according to TERRA content (Supplementary Fig. 3c). Nevertheless, RNAse H1 exposure led to significantly weaker TERRA binding compared to a mock-treated sample, implying that the BRCA1–TERRA interaction is predominantly R-loop dependent (Fig. 2e). RNAse A treatment confirmed RNA-specific signals (Fig. 2e). BRCA1–TERRA binding was also confirmed in HME cells (Supplementary Fig. 3d).

Consistent with observations that ALT cells are enriched in TERRA R-loops due to elevated TERRA expression by comparison with telomerase-positive cells[9], we also detected greater R-loop-associated BRCA1–TERRA colocalization in cells with higher levels of TERRA/telomeric R-loops, i.e., in U2OS than in HME cells (Fig. 2f and Supplementary Fig. 3e, f). Similar results were obtained with BRCA1 occupancy at telomeres, indicating that ALT cells may rely on the R-loop-based BRCA1–TERRA functionality to maintain telomere integrity (Supplementary Fig. 3g, h).

Interestingly, we observed colocalized BRCA1–TERRA solely in PML bodies, a subtype of which, ALT-associated APBs, are important for the repair of damaged ALT telomeres and ALT DNA synthesis[34] (Supplementary Fig. 3i, j). Since the abundance of colocalized foci decreased upon treatment with RNAse H1, we suggest that BRCA1–TERRA interactions in APBs are driven by damaged ALT telomeres and/or ALT DNA synthesis, since both cases are likely to involve TERRA R-loops[9].

**BRCA1 association with TERRA is dynamic throughout the cell cycle.** Considering the cell cycle-dependent expression of both BRCA1 and TERRA RNA[35,36], we examined the dynamics of BRCA1–TERRA binding by performing a BRCA1 RIP using chromatin fractions from synchronized T98G and HME cells (Fig. 2g–i and Supplementary Fig. 3k–m).

Consistent with others[35], we observed that TERRA levels peaked in G1, decreased through S phase, and rose again in G2 (Fig. 2h and Supplementary Fig. 3l, see input lanes). We also observed dynamic BRCA1–TERRA binding on chromatin, with the first peak appearing in early S and the second in late S–G2/M,

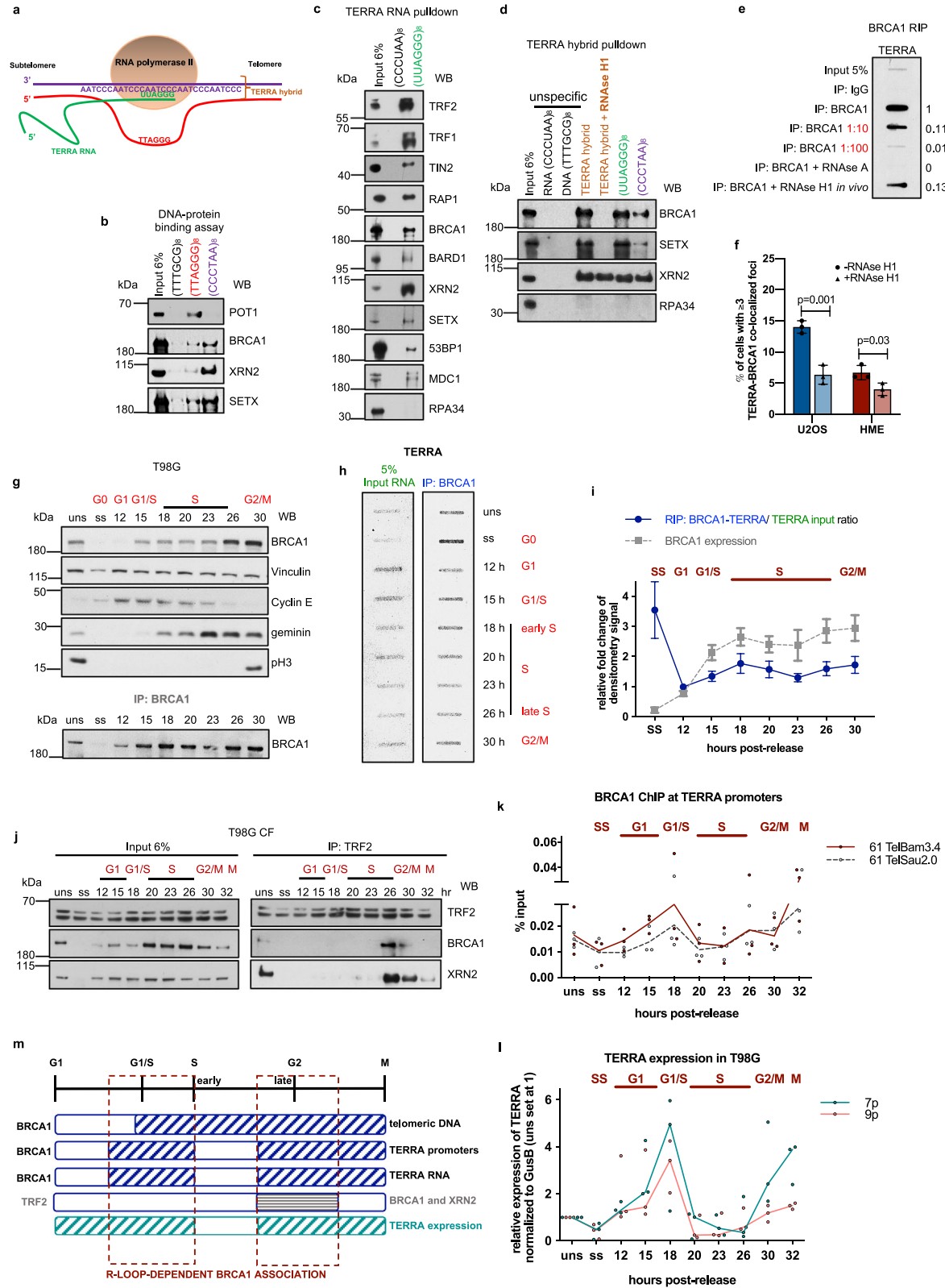

respectively (Fig. 2h, i and Supplementary Fig. 3l, m). Furthermore, exposure to RNAse H1 in HME cells significantly diminished BRCA1–TERRA binding in all phases of the cell cycle, with a particular decline in S and G2 (Supplementary Fig. 3n, o). Besides G1, little interaction with TERRA was detected in mid S phase (23 h in T98G and 18 h in HME; Fig. 2h, i and Supplementary Fig. 3l, m), possibly due to low TERRA expression

in this particular time frame, since abundant TERRA likely hampers telomeric DNA replication[2,37].

Human telomere replication timing occurs throughout S phase with the spike in late S/G2, important for unfolding the T-loop structure and for efficient replication termination[38,39]. Moreover, TERRA itself was found at T-loop junctions, likely adding to their stability, perhaps by forming an internal R-loop together with

**Fig. 2 BRCA1 association with TERRA is predominantly R-loop-, and cell cycle-dependent. a** Schematic representation of TERRA synthesis. **b, c** DNA- (**b**) or RNA- (**c**) binding assay using biotinylated DNA or RNA oligonucleotides, which were incubated with HeLa CF and analyzed by immunoblotting with the indicated antibodies. **d** TERRA hybrid-binding assay using biotinylated RNA (UUAGGG)$_8$ and DNA (CCCTAA)$_8$ oligonucleotides. Hybrids were mock- or RNAse H1-treated prior to the incubation with HeLa CF, and analyzed by immunoblotting with the indicated antibodies. **e** A BRCA1 RIP experiment was performed with mock- or RNAse H1-treated U2OS cells, followed by DNAse treatment of the recovered RNA and a slot blot, using CCCTAA-DIG probe. **f** A mean percentage of ≥3 positive BRCA–TERRA colocalization events from IF-FISH experiments in mock- or RNAse H1-treated U2OS and HME cells from three independent experiments. At least $n = 100$ cells scored/experiment, ±SD. P values were obtained using a two-tailed Student's $t$ test. **g** An immunoblot of T98G cells confirming the efficiencies of synchronization and the BRCA1 RIP assay. **h** The BRCA1 RIP-slot blot analysis performed in synchronized T98G depicted in **g**. RNA isolated from each BRCA1 IP was treated with DNAse, and TERRA was detected by hybridization with CCCTAA-DIG probe. **i** Densitometry analysis of the BRCA1 RIP experiment in **h**. The panel represents the mean ratio of relative fold changes of TERRA RIP signal values over TERRA input signal values, $n = 3$ independent experiments, ±SD. The relative BRCA1 expression in this experiment, as depicted in **g**, is shown in gray. **j** Cell cycle-dependent interaction of endogenous TRF2 with BRCA1 and XRN2. CFs of synchronized T98G cells were IPd with TRF2 antibody prior to immunoblotting. $n = 3$ independent experiments. **k** BRCA1 ChIP analyses of TERRA promoter regions performed in synchronized T98G cells. **l** TERRA expression level from 7p and 9p subtelomeric regions in synchronized T98G cells. Values were normalized to GusB and compared to unsynchronized sample, set at 1. **k, l** Each dot represents a biological replicate ($n = 3$), and mean (**k**) or median (**l**) values are connected by a line. **m** A schematic drawing of cell cycle-dependent BRCA1 interactions with TERRA, TERRA promoters, telomeric regions, TRF2, and XRN2. Source data are provided as a Source data file.

---

TRF2[40–42]. Therefore, the two R-loop-dependent peaks of BRCA1–TERRA binding could be related to BRCA1-mediated facilitation of telomere replication, when TERRA expression is high enough, in G1/S-early S and late S/G2 phases, as well as to the unwinding of the T-loop to successfully complete replication. Interestingly, we also observed robust binding of BRCA1 and XRN2 to TRF2 on chromatin in these particular phases, late S and G2 (Fig. 2j and Supplementary Fig. 3p). Furthermore, XRN2 association with TRF2 is likely to be BRCA1-dependent (Supplementary Fig. 3q). Thus, these data suggest the possible formation of functional complexes that include BRCA1, XRN2, TRF2, TERRA RNA, and possibly other proteins engaged in unfolding T-loops and/or stabilizing them once telomere replication is complete.

Surprisingly, we also noticed that TERRA strongly associated with BRCA1 in serum-starved (SS), i.e., quiescent, HME and T98G cells, despite the very low TERRA and BRCA1 levels in quiescent compared to cycling cells (Fig. 2h, i and Supplementary Fig. 3l, m; see SS lanes of input and IP). This suggests that BRCA1–TERRA interactions that arose amidst the low TERRA and BRCA1 expression levels in G0 cells perform an unexpected and yet unknown function during this period.

Finally, we also detected cell cycle-dependent BRCA1 binding to CpG-rich TERRA promoters, which occurred in a pattern similar to that of TERRA expression, suggesting that BRCA1 also controls TERRA expression timing (Fig. 2k, l and Supplementary Fig. 3p).

Overall, these results consistently reflect a cell cycle-, and an R-loop-dependent association of BRCA1 with TERRA and its promoters, telomeric regions, TRF2, and XRN2, possibly contributing to proper telomeric DNA replication (Fig. 2m).

**BRCA1 regulates TERRA expression and TERRA R-loop abundance.** Since our results imply that BRCA1 participates in the resolution of TERRA R-loops and binds to TERRA promoters (Figs. 1g–k and 2k), we next measured TERRA expression in the presence and absence of BRCA1. Slot blot analyses revealed that BRCA1 depletion led to elevated TERRA levels in all three cell lines being tested, with U2OS exhibiting the most dramatic effect (Fig. 3a, b and Supplementary Fig. 4a). Moreover, TERRA levels decreased upon RNAse H1 overexpression, suggesting that more abundant TERRA is associated with an elevated amount of TERRA R-loops in BRCA1-deficient cells (Fig. 3a, b). An increased TERRA abundance was also observed at chromosome ends in metaphase spreads of BRCA1-depleted HME cells (Supplementary Fig. 4b, c).

We next assessed TERRA levels on individual chromosomes by qRT-PCR in the same cell lines using a panel of subtelomere-specific primers (Fig. 3c). Consistently, BRCA1 depletion triggered an upregulation of TERRA expression at the majority of chromosome ends tested and its consequent decrease following RNAse H1 exposure (Supplementary Fig. 4d, e). Interestingly, TERRA levels varied among the different subtelomeric regions in the cell lines tested (Fig. 3c), which could be a product of the cell type, sequence characteristics of the subtelomeric regions, GC-skew, and/or alterations in relevant chromatin structure.

Furthermore, we observed increased TERRA expression in BRCA1$^{+/-}$ cells and an even greater upregulation in HME BRCA1$^{-/-}$ cells (Fig. 3d). Considering the prominent genomic instability, telomere dysfunction, and premature senescence reported in HME cells from BRCA1 mutation carriers (BRCA1$^{mut/+}$)[19], we speculate that elevated TERRA expression and formation of TERRA R-loops in BRCA1$^{+/-}$ cells is a manifestation of BRCA1 haploinsufficiency. In this setting, defective BRCA1 function may have led to accumulating DNA damage at telomeres (Fig. 1j).

Depletion of SETX or XRN2 also resulted in elevated total TERRA expression, which was downregulated following RNAse H1 treatment (Supplementary Fig. 4f, g). Moreover, changes in chromosome-specific TERRA abundance developed consistently in the absence of SETX or XRN2 or BRCA1 (Supplementary Fig. 4h, i). These results, together with our previous findings (Figs. 1e and 2b–d), suggest the existence of an interplay between BRCA1, SETX, and XRN2, leading to the regulation of TERRA abundance and TERRA R-loops.

**BRCA1 regulates TERRA transcription.** TERRA upregulation in BRCA1-depleted cells is likely not the result of the minimal variations observed in either TERRA transcripts' half-life or cell cycle perturbations (Supplementary Fig. 4j, k). Since increased TERRA levels in BRCA1-deficient cells correlate with increased R-loop abundance (Figs. 1g and 3a, b), which likely develops co-transcriptionally[43], we compared the TERRA transcription profiles in control and BRCA1-depleted cells. Specifically, we performed ChIP experiments using antibodies directed at RNAPII and its two phosphorylated forms, RNAPII-S5P and RNAPII-S2P, which initiate and elongate/terminate transcription, respectively[44] (Fig. 3e, f).

ChIP slot blot analyses revealed that BRCA1 depletion led to elevated binding of all three RNAPII forms to telomeric DNA, which implies an increase in transcription (Fig. 3e, f). We also observed transcription upregulation in subtelomeric regions in

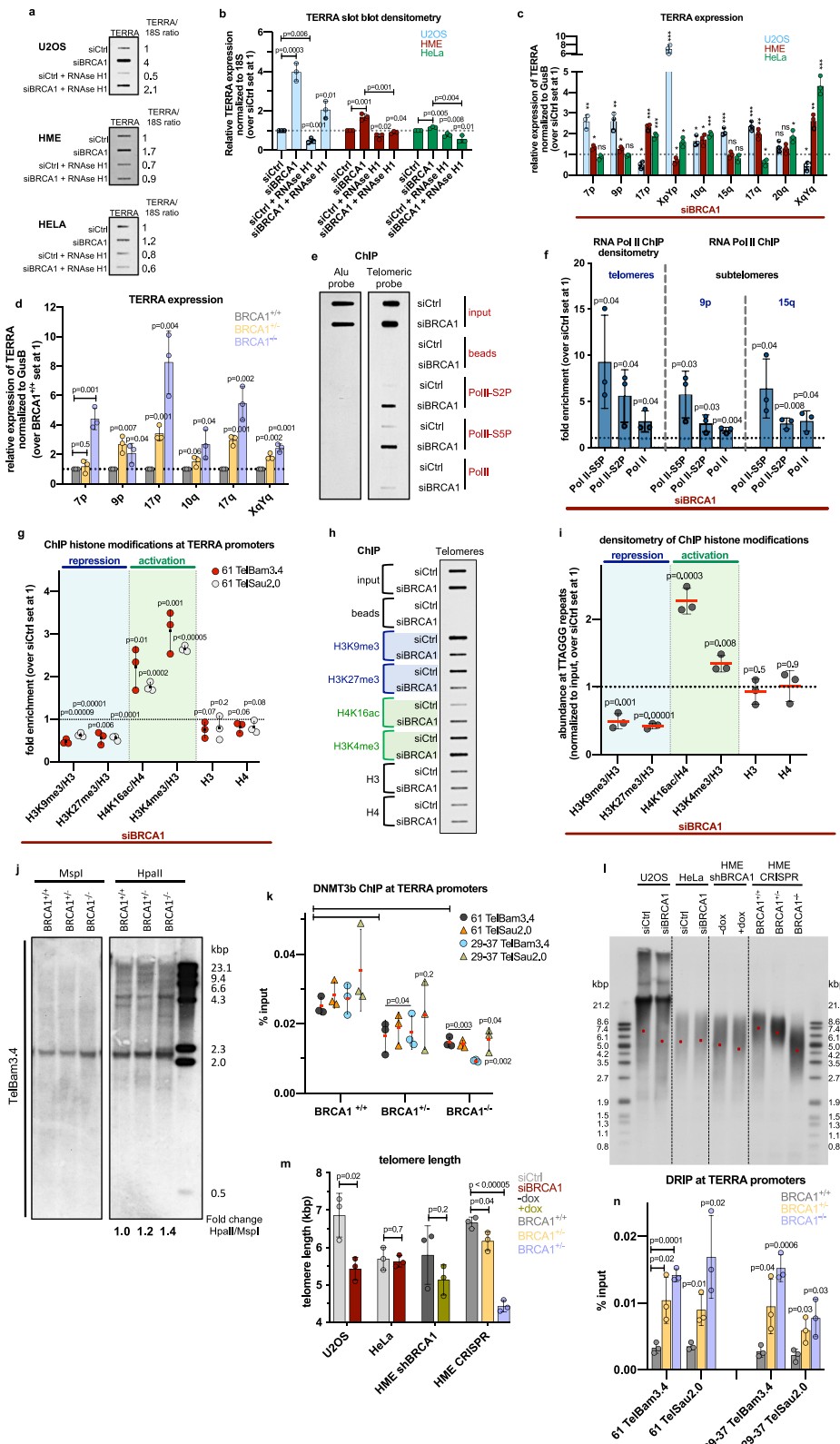

BRCA1-depleted cells, consistent with enhanced TERRA transcription initiation (Fig. 3f).

Increased RNAPII binding to subtelomeric and telomeric regions in BRCA1-depleted cells could result from altered chromatin architecture at TERRA promoters at the level of histone modifications and/or DNA methylation. Thus, we assessed the relative abundance of H3K9me3, H3K27me3,

H4K16ac, and H3K4me3 marks at TERRA promoters, with the first two being repressive marks and the latter two activating ones. Relative densities of H3K9me3 and H3K27me3 were decreased following BRCA1 depletion, whereas the H4K16ac and H3K4me3 occupancies were increased (Fig. 3g). Furthermore, a similar pattern of histone-mark distribution was observed at telomeric regions[45] (Fig. 3h, i), indicating an open chromatin conformation

**Fig. 3 BRCA1 depletion results in upregulated TERRA transcription. a, b** Slot blot and respective densitometry analyses of total TERRA levels in control and BRCA1-depleted samples, ±RNAse H1 exposure. Bars represent the average value of TERRA signals, normalized to 18S, ±SD. **c** qRT-PCR analysis of TERRA levels from individual subtelomeres in control and BRCA1-depleted U2OS, HME, and HeLa cells. The data were normalized to GusB and compared to siCtrl, set at 1, ±SD, *$p < 0.05$, **$p < 0.005$, ***$p < 0.0005$ (two-tailed Student's *t* test). **d** TERRA levels in CRISPR-modified HME cells from different subtelomeres, normalized to GusB and compared to BRCA1$^{+/+}$ values, ±SD. **e** ChIP slot blot of telomeric DNA and Alu repeats with anti-RNAPII antibodies (RNAPII, RNAPII-S5P, and RNAPII-S2P) performed in control and BRCA1-depleted U2OS cells. $n = 3$ independent experiments. **f** Densitometry analysis of ChIP experiments in **e** (left), and qRT-PCR of 15q and 9p subtelomeric amplicons from the same ChIP experiments (right). Data are represented as fold change over control sample, set at 1, ±SD. **g** ChIP analyses of promoter regions performed in control and BRCA1-depleted U2OS cells using the indicated antibodies. Data are represented as fold enrichment over siCtrl. Values were normalized to H3 or H4, ±SD. **h** ChIP slot blot of telomeric DNA performed in control and BRCA1-depleted U2OS cells, using the same antibodies as in **g**. $n = 3$ independent experiments. **i** Densitometry analysis of ChIP experiments in **h**. Data are represented as fold change over siCtrl, ±SD. **j** TERRA CpG-island promoter methylation analysis of CRISPR-modified HME cells. Genomic DNA was digested with the methylation-insensitive MspI or methylation-sensitive HpaII. DNA was hybridized using a DIG-labeled probe detecting TERRA promoter CpG-island repeats. $n = 3$ independent experiments. **k** ChIP analyses of promoter regions performed in CRISPR-modified HME cells using DNMT3b antibody, ±SD. **l, m** Telomere length analysis. Genomic DNA was digested with HinfI and RsaI restriction enzymes and subjected to Southern blot using DIG-labeled CCCTAA probe, ±SD. **n** DRIP analyses of promoter regions performed in CRISPR-modified BRCA1 HME cells, ±SD. For all quantitation plots (**b**, **d**, **f**, **g**, **i**, **k**, **m**, **n**; $n = 3$ independent experiments) *p* values were computed using a two-tailed Student's *t* test. Source data are provided as a Source data file.

at TERRA promoters and telomeres, which could, in turn, facilitate RNA PolII recruitment. Similarly, enriched telomeric H4K16ac was detected in HME cells from BRCA1 mutation carriers[19], again suggesting that increased TERRA expression and R-loop formation serve as a route to telomeric dysfunction.

Furthermore, we observed hypomethylation of TERRA CpG-island promoters in CRISPR-modified BRCA1$^{+/-}$ and BRCA1$^{-/-}$ HME and in transiently BRCA1-depleted U2OS cells, as demonstrated by an increased heterogeneity of hybridization patterns detected by methylation-specific Southern blot (Fig. 3j and Supplementary Fig. 4l). Since demethylation of CpG-island promoters has been linked to a deficiency of DNMT1 and DNMT3b and derepressed TERRA synthesis[5,37], we assessed their recruitment to TERRA promoters in a BRCA1-depleted setting (HME and U2OS) and found diminished occupancy of DNMT3b, but not DNMT1 in these regions (Fig. 3k and Supplementary Fig. 4m, n).

Enhanced TERRA transcription and TERRA R-loop formation have been linked to telomere shortening as a result of increased telomeric DNA damage[9,37]. Consistently, we observed significant telomere loss in cell lines exhibiting high TERRA levels in the absence of BRCA1, i.e., in U2OS and CRISPR-modified BRCA1 HME cells (Fig. 3l, m and Supplementary Fig. 4o).

Interestingly, in CpG-island-containing promoters, R-loop formation negatively correlates with DNA methylation and thus reflects transcription activation[46]. Since we observed R-loop-dependent binding of BRCA1 to CpG-rich TERRA promoters (Fig. 1g) and increased R-loop-generated TERRA expression in the absence of BRCA1 (Supplementary Fig. 4d), we propose that BRCA1 regulates the TERRA production through suppression of R-loop formation at TERRA promoters. Depletion of BRCA1 results in elevated R-loop levels at promoter regions (Fig. 1h), accompanied by associated histone-mark distribution changes and reduced recruitment of DNMT3b. Demethylated promoters become accessible to RNA PolII binding, leading to elevated TERRA transcription, accompanied by upregulated R-loops in telomeric regions, which in turn result in telomere DNA damage and shortening.

Thus, shorter telomere length in BRCA1$^{-/-}$ cells compared to BRCA1$^{+/-}$ cells could be caused by higher levels of R-loops at TERRA promoters in BRCA1$^{-/-}$ cells (Fig. 3n), followed by overly increased TERRA expression (Fig. 3d), resulting in shortened telomere length and consequently exacerbated TERRA abundance[45].

**BRCA1 directly interacts with the TERRA UUAGGG repeats.** Based on our RIP results (Fig. 2e–h), we next asked whether there

are discrete regions in the BRCA1 polypeptide that mediate TERRA binding. We employed six, overlapping GST-tagged BRCA1 fragments[31] and tested their abilities to bind TERRA repeats in the absence and presence of added cellular chromatin lysate (Fig. 4a, b and Supplementary Fig. 5a). The TERRA pull-down revealed that GST-F3 exhibited the strongest binding to TERRA, and a less prominent interaction was detected with GST-F2 (Fig. 4b and Supplementary Fig. 5a). Surprisingly, the BRCA1–TERRA interaction seemed to be direct, given its occurrence in the presence of lysate-free GST-BRCA1 fragments only (Fig. 4b). Since GST-F2 and GST-F3 contain a ~50 aa overlap in their C-, and N-termini, respectively, and since GST-F3 revealed stronger binding to a TERRA oligonucleotide than GST-F2 (Fig. 4b), we focused on the former to further define the segment(s) of BRCA1 important for TERRA interaction.

We truncated GST-F3 into subfragments and found that GST-F3.1.1 (500–510 aa) was responsible for TERRA binding (Fig. 4c and Supplementary Fig. 5b). Interestingly, GST-F3.1.1 contains a basic motif with the sequence KRKRR, which corresponds to the most N-terminal of the two BRCA1 nuclear localization signals (NLS1; 503–507 aa)[47]. Intriguingly, several RNA processing proteins, such as U1-70K and Dicer, use their NLSs as RNA-binding domains, in part due to their high positive charge density, which is favorable for RNA binding[48–50].

We, therefore, asked whether deletion of BRCA1 NLS1 abrogates BRCA1 ability to interact with TERRA. In this context, a cancer-associated mutation (lung adenocarcinoma) c. 1518 G>T (R506S) was identified within the BRCA1 503–507 aa segment (cancer.sanger.ac.uk). Due to the extreme N-terminal position of the KRKRR motif in GST-F3, we introduced the ΔNLS1 and R506S mutations into BRCA1 GST-F2, which also contains the NLS1 sequence. A TERRA pulldown revealed that, unlike GST-F2-WT, the GST-F2-ΔNLS1 mutant no longer bound TERRA, and GST-F2-R506S exhibited a significantly diminished interaction with TERRA (Fig. 4d).

Notably, we did not observe TERRA binding to the NLS2-containing (607–614 aa) GST-BRCA1 fragment GST-F3.2 (Supplementary Fig. 5b).

Taken together, these data indicate the importance of the KRKRR motif for direct BRCA1–TERRA binding in vitro.

**BRCA1 defect in TERRA binding leads to abundant telomeric R-loops.** To further evaluate the impact of the ΔNLS1 and R506S mutations on BRCA1 function, we generated full length HA-tagged BRCA1 vectors harboring the aforementioned sequence alterations. The relevant vectors exhibited efficient expression and

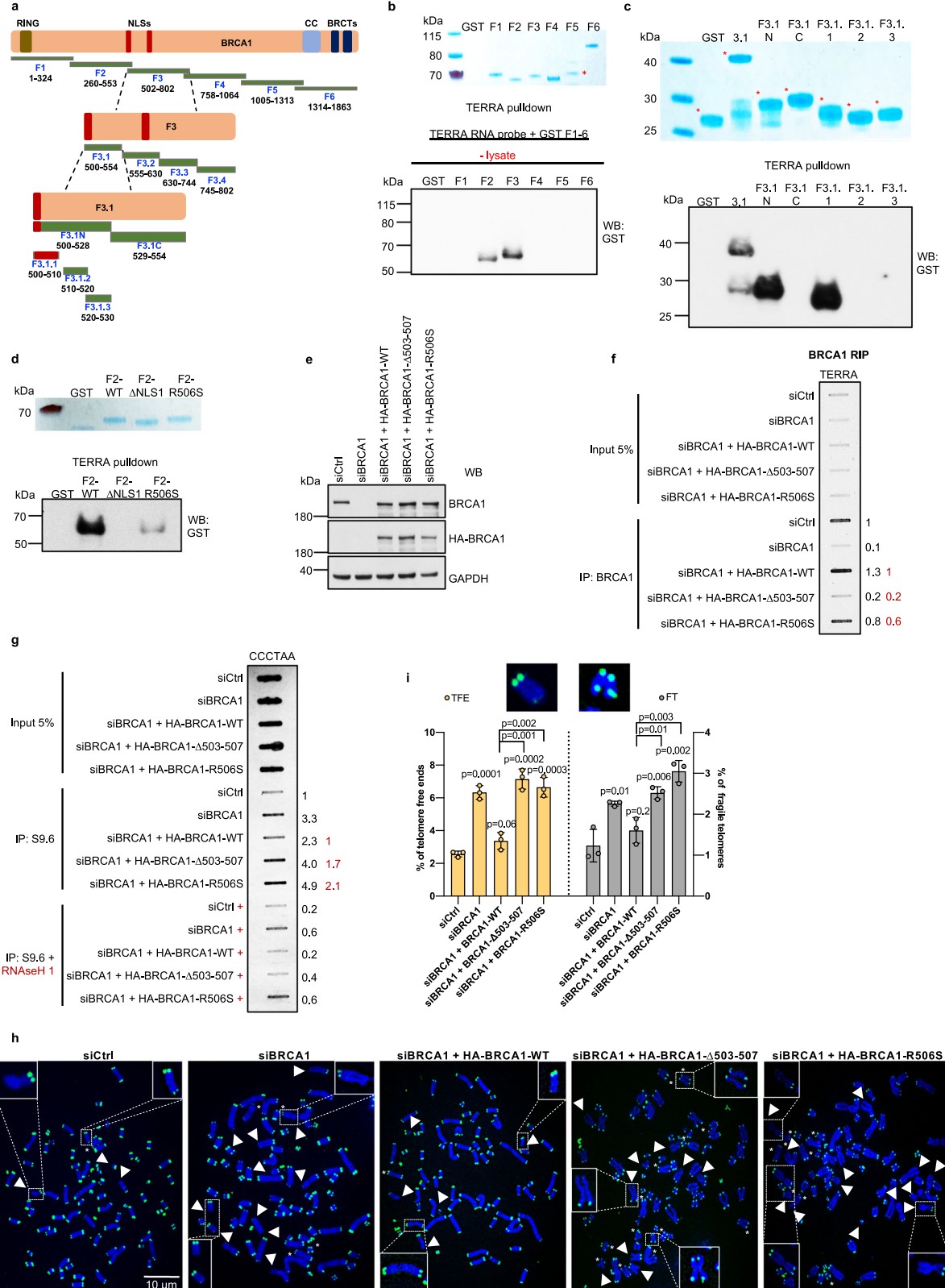

nuclear localization despite a lack of the NLS1 (Supplementary Fig. 5c, d).

To examine binding of BRCA1 mutants to endogenous TERRA, U2OS cells were depleted of BRCA1 using a 3′-UTR-targeted siRNA and further transfected with WT-BRCA1 or each relevant mutant (Fig. 4e). A subsequent RIP assay revealed that both BRCA1 mutants exhibited diminished interactions with TERRA, with the ΔNLS1 mutant revealing a stronger binding defect than the point mutant (Fig. 4f).

We next analyzed the abundance of telomeric R-loops generated by the BRCA1 mutants by performing DNA–RNA immunoprecipitation (DRIP), followed by a slot blot with a probe complementary to the R-loop-generating DNA strand. BRCA1 depletion resulted in a 3.3-fold increase in telomeric hybrid

**Fig. 4 TERRA binding-associated BRCA1 mutants display abundant telomeric R-loops and telomere dysfunction. a** Schematic representation showing the sequence-based architecture of BRCA1 and the GST fragments used in this study. **b**, **c** TERRA pulldown assays, using BRCA1 GST fragments. A biotinylated TERRA probe was incubated with the GST-BRCA1 fusion proteins described in **a** in the absence of lysate. The TERRA-bound GST-BRCA1 fragment complexes were analyzed by immunoblotting using anti-GST antibody. Coomassie blue staining revealed the relative abundance of GST fusion proteins. **d** TERRA affinity assay using BRCA1 GST-F2-WT, GST-F2-ΔNLS1, and GST-F2-R506S fragments. A biotinylated TERRA probe was incubated with GST-BRCA1 fusion proteins only, and the pulldown was analyzed by immunoblot using anti-GST antibody. **e** Western blot of U2OS cells used in **f** and **g** confirming depletion of endogenous BRCA1, and expression of HA-tagged WT and mutant BRCA1. **f** Interaction of ΔNLS1 and R506S BRCA1 mutants with TERRA. A BRCA1 RIP assay was performed in U2OS cells depleted of endogenous BRCA1 with subsequent transfection of WT and mutant BRCA1. Recovered RNA was treated with DNAse, and TERRA was detected by a slot blot. Densitometry analysis was assessed, and the value of siCtrl (in black) and WT-BRCA1 (in red) were set at 1. **g** DRIP-slot blot analysis of U2OS cells depleted of BRCA1 with subsequent transfection of WT and mutant BRCA1 vectors. Extracted DNA was mock- or RNAse H1-treated, and subjected to DRIP-slot blot, using a probe complementary to the CCCTAA repeats. Densitometry analysis was performed, and the values of siCtrl (in black) and WT-BRCA1 (in red) were set at 1. **b**–**g** $n = 3$ independent experiments. **h** Representative images of FISH experiments performed in metaphase spreads from U2OS cells depleted of endogenous BRCA1, and transfected with WT and mutant BRCA1 vectors. Telomeric DNA was stained in green, while DAPI-stained chromosomes are in blue. Arrowheads point to telomere-free ends (TFEs) and asterisks to fragile telomeres (FTs), respectively. **i** Quantitation of TFEs and FTs in metaphase spreads as depicted in **h**. Graphs represent the mean percentage of indicated telomeric aberrations/metaphase from three independent experiments. At least $n = 30$ metaphases scored/experiment, ±SD. *P* values were obtained using a two-tailed Student's *t* test. Source data are provided as a Source data file.

formation as compared to control cells, with a partial rescue of TERRA R-loop abundance upon transfection with WT-BRCA1 (Fig. 4g). Reconstitution with either BRCA1 mutant led to the upregulated hybrid formation at telomeres, similar to that observed in BRCA1-depleted cells (Fig. 4g).

In contrast, R-loop levels at TERRA promoters and TERRA expression in mutant BRCA1-expressing cells were significantly diminished compared to BRCA1-deficient cells (Supplementary Fig. 5e, f). This suggests that these BRCA1 mutants, defective in binding to TERRA UUAGGG repeats, are still able to efficiently suppress TERRA abundance, perhaps at the level of R-loop-mediated transcription.

In keeping with these results, binding of the BRCA1 mutants to CpG-rich TERRA promoters remained unchanged (Supplementary Fig. 5g, h). However, we observed increased binding of these mutants to telomeric DNA repeat sequences, perhaps as a result of their retention on telomeric chromatin due to unresolved telomeric R-loops (Supplementary Fig. 5g, h and Fig. 4g). Presumably, as a result of accumulated telomeric R-loops, we observed elevated telomeric aberrancies, such as telomere-free ends and fragile telomeres, in the BRCA1 mutant-expressing U2OS cells (Fig. 4h, i and Supplementary Fig. 5i).

Interestingly, a similar phenotype was observed in RNAse H1-depleted ALT cells[9]. Since precise levels of TERRA hybrids are important for maintaining telomeric HR in ALT cells, we hypothesize that BRCA1-deficient cells, as well as TERRA binding-defective BRCA1 mutants, fail to support telomeric HR due to an abnormally high accumulation of unresolved telomeric R-loops (Fig. 4g–i).

However, unlike wt BRCA1 depletion, neither the ΔNLS1 nor the R506S exhibited a significant defect in non-telomeric HR (Supplementary Fig. 5j). Thus, we assume that non-telomeric HR operates by different rules from telomeric HR, and TERRA–BRCA1 binding appears to be critical in the latter, but not the former.

Altogether, our findings suggest the biological importance of the BRCA1 region responsible for TERRA binding.

**BRCA1 suppresses replication stress arising from TERRA R-loops.** TERRA R-loops could compromise telomere stability by impairing telomere replication[9,10]. Thus, we next asked whether BRCA1 depletion-triggered TERRA upregulation causes replication stress at telomeres.

IF-FISH experiments revealed a BRCA1 depletion-dependent increase of colocalized pATR[Thr1989] and pChk1[S345], reporters of defective replication[51,52], with TERRA RNA at telomeres (defined by TRF2 co-staining; Fig. 5a–c and Supplementary Fig. 6a). Moreover, these colocalizations primarily co-occurred with PML bodies, an increased abundance of which in the absence of BRCA1 is also suggestive of telomere region-specific replication stress[53] (Supplementary Fig. 6b–f). RNAse H1 exposure further diminished pATR/pChk1-TERRA colocalization at telomeres, suggesting that elevated TERRA R-loops are likely to be the cause of increased telomeric replication stress. Furthermore, we observed elevated replication defects in cells expressing the ΔNLS1 and the R506S BRCA1 mutants, suggesting an importance of BRCA1–TERRA interaction in suppression of replication-associated telomeric damage (Fig. 5d, e and Supplementary Fig. 6g–i).

To investigate the consequences of this effect, we further analyzed the "fitness" of telomere ends. We performed telomere DNA FISH in metaphase spreads of four cell lines: non-immortalized (no hT) and immortalized (hT) HME, U2OS, and HeLa cells (Fig. 5f, g). BRCA1 depletion led to an accumulation of telomeric abnormalities in all of these cells with the exception of HeLa, the cell line with the smallest increase of TERRA levels in the absence of BRCA1 (Figs. 3a and 5f, g). Thus, one might speculate that upregulated TERRA levels in BRCA1-deficient cells result in an increase of unresolved telomeric R-loops, leading to replication stress and subsequent telomeric dysfunction, potential hallmarks of forthcoming tumourigenesis[54].

## Discussion

Tight regulation of TERRA abundance appears to be essential for telomeric integrity, since either its depletion or its abnormal upregulation leads to elevated telomeric DNA damage[10,33]. Recent reports have implicated TERRA in the formation of DNA damage-generating, telomeric RNA–DNA hybrids[9,10]. Consistent with this, we have demonstrated that BRCA1 associates with the main members of the shelterin complex and with TERRA. The formation of these biochemical complexes seems to occur predominantly in an R-loop-associated manner at promoter and telomeric regions. In this context, BRCA1 limits TERRA levels in order to modulate the persistence of TERRA R-loops at telomeres, which, when uncontrolled, can endanger telomere stability.

Specifically, we find that BRCA1-mediated suppression and/or resolution of R-loops at subtelomeric and telomeric regions modulates their heterochromatic state, including methylation of CpG-island TERRA promoters and enrichment of repressive histone marks H3K9me3 and H3K27me3 (Fig. 6). BRCA1 depletion results in an impaired recruitment/binding of DNMT3b

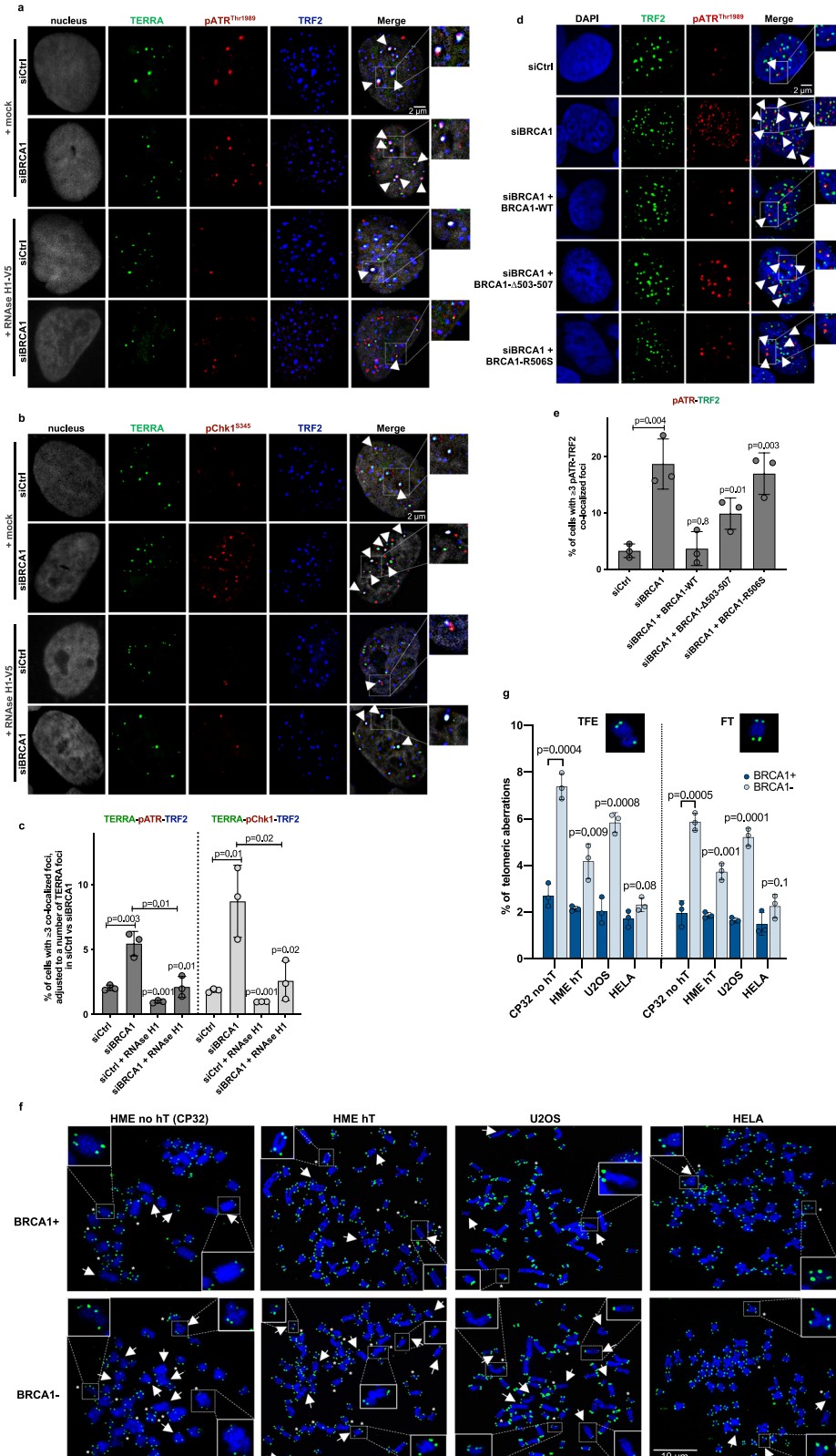

to CpG-rich TERRA promoters, which leads to their demethylation. We believe that this phenomenon is the consequence of TERRA R-loop accumulation at these regions, however, we cannot exclude that BRCA1 may also directly promote DNMT3b binding to TERRA promoters.

Promoter demethylation is accompanied by the increased abundance of activating histone marks H4K16ac and H3K4me3,

resulting in RNA PolII recruitment and subsequent upregulated transcription (Fig. 6). Alternatively, BRCA1 depletion-triggered elevated TERRA transcription could be a result of increased binding of transcription factors to TERRA type II promoters, devoid of CpG-islands[55,56]. Excessive TERRA levels together with an accumulation of unresolved TERRA R-loops at telomeric regions lead to replication stress and telomeric aberrancies,

**Fig. 5 BRCA1 suppresses replication stress at telomeres arising from TERRA R-loops. a, b** Representative images of TERRA RNA-FISH combined with anti-TRF2 and anti-pATR$^{Thr1989}$ (**a**) or anti-pChk1$^{S345}$ (**b**) immunostaining performed in mock- or RNAse H1-treated siCtrl and BRCA1-depleted U2OS cells. Triple colocalization events are depicted with arrowheads. **c** Mean percentage of ≥3 TERRA-TRF2-pChk1$^{S345}$/pATR$^{Thr1989}$ colocalization events per nucleus, adjusted to a number of TERRA foci in control and BRCA1-depleted cells, from three independent experiments. At least $n = 60$ cells scored/experiment, ±SD. P values were computed using a two-tailed Student's $t$ test. **d** Representative images of combined immunofluorescence using anti-TRF2 and anti-pATR$^{Thr1989}$ immunostaining performed in U2OS cells, depleted of endogenous BRCA1, and transfected with a WT or a mutant BRCA1 vector. Colocalization events are depicted with arrowheads. **e** Mean percentage of ≥3 TRF2-pATR$^{Thr1989}$ colocalized foci depicted in **d** from three independent experiments. At least $n = 80$ cells scored/experiment, ±SD. P values were calculated using a two-tailed Student's $t$ test. **f** Examples of FISH experiments performed in metaphase spreads from HME ± hT, U2OS, and HeLa cells transfected with siCtrl or siBRCA1. Telomeric DNA is in green and DAPI-stained chromosomes in blue. Arrows point to telomere-free ends (TFEs) and asterisks to fragile telomeres (FTs), respectively. **g** Quantitation of TFEs and FTs in metaphase spreads of FISH experiments performed in **f**. Graphs represent the mean percentage of indicated telomeric aberrations/metaphase from three independent experiments, $n = 30$ metaphases scored/experiment, ±SD. P values were computed using a two-tailed Student's $t$ test.

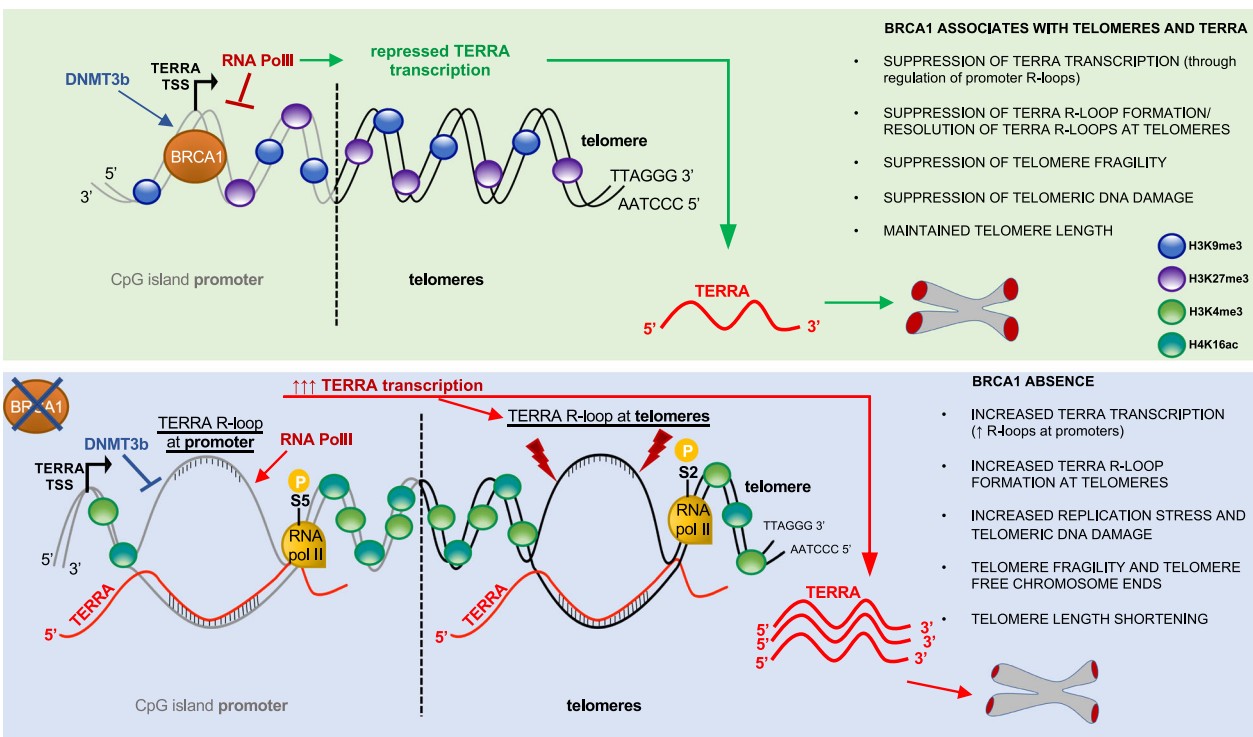

**Fig. 6 Model of TERRA-associated BRCA1 functions at telomeres.** Schematic representation of BRCA1 function at telomeres and consequences of BRCA1 depletion on TERRA levels and telomere integrity.

resulting in telomere shortening (Fig. 6). Telomere length reduction in turn forms a feedback loop by exacerbating the upregulation of TERRA expression[45].

BRCA1-associated TERRA R-loop regulation and/or resolution is likely performed in collaboration with SETX and XRN2. In support of this hypothesis, we observed (a) strong binding of both of these proteins to TERRA and to TERRA hybrids, (b) total and chromosome-specific, increased TERRA expression, partially caused by elevated R-loops, when any of these proteins is absent, and (c) increased telomeric DNA damage in XRN2-deficient cells.

Interestingly, we also observed BRCA1 haploinsufficiency for the regulation of R-loop-mediated TERRA expression and for the suppression of R-loop-associated telomeric DNA damage in our CRISPR-mutated HME cells, suggesting that both defects are candidate contributors to tumourigenesis in BRCA1-deficient cells.

We also find that BRCA1 can directly interact with TERRA via its NLS1 region. However, we cannot rule out the possibility that additional, currently unidentified BRCA1 polypeptide sequences modulate TERRA binding. We have also demonstrated that cells harboring mutations within the TERRA-interacting region of

BRCA1, ΔNLS1 and the clinically relevant R506S, (a) have a BRCA1/TERRA binding defect, (b) exhibit elevated TERRA-nucleated R-loops, and (c) display replication stress and telomeric abnormalities, emphasizing the importance of a normal BRCA1–TERRA association in the maintenance of telomere integrity and prevention of tumourigenesis.

Our work also reveals a cell cycle-dependent interaction of BRCA1 with TERRA and its promoters, which peak both in early S and in late S–G2/M. This suggests that BRCA1 ensures proper telomere replication by regulating the formation and/or the repair of TERRA R-loops, and perhaps by the unfolding of T-loops to replicate chromosomal extremities.

Surprisingly, BRCA1 and TERRA also appear to interact in quiescent cells. To date, there are no reports of TERRA levels/transcription in quiescent human cells. However, recent studies in fission yeast indicate that TERRA transcription in G0 is even higher than in cycling cells, thereby promoting rearrangements of eroded telomeres and preventing cells from re-entering the cell cycle[57]. Similarly, quiescent hematopoietic stem cells with short telomeres manifested a greater tolerance for accumulating genomic alterations, but failed to transmit them to progenitor cells as a

result of having undergone apoptosis or senescence[58]. Thus, we suggest that BRCA1 prevents the emergence of DNA damage at telomeres, in part, by interacting with TERRA in quiescent cells and, conceivably, ensuring their successful entry into the cell cycle.

Finally, we speculate that our results might be relevant to potential telomere-based cancer therapies. Since targeting R-loops has been considered as a tumor therapy approach, an excess of TERRA R-loops in BRCA1-deficient cells represents a potential target for R-loop-directed therapy for BRCA1 mutation carriers. Interestingly, tumors harboring BRCA1/2 mutations are highly sensitive to trabectedin and lurbinectedin[59,60], compounds that induce R-loop-driven damage and cause genomic instability[61]. Thus, R-loop-targeting drugs, possibly in combination with a telomerase inhibitor, might represent interesting approaches to BRCA1-mutant tumor therapy.

## Methods

**Cell lines**. HeLa and U2OS cells were grown in 10% FBS-containing DMEM medium supplemented with 50 μg/ml penicillin–streptomycin (Gibco). T98G cells were maintained in DMEM supplemented with 10% FBS, 50 μg/ml penicillin–streptomycin, and 1% glutamine (Gibco). Non-immortalized and telomerase immortalized HME were maintained in MEGM medium (PromoCell). CRISPR-modified HME BRCA1[+/+], BRCA1[+/−], and BRCA1[−/−] cell lines were cultured in MEGM medium. hT HME containing tetracycline-inducible shBRCA1 were grown in MEGM medium containing 500 ng/ml doxycycline[32]. All cells were cultivated in a humidified incubator at 37 °C in a 5% CO$_2$-containing atmosphere.

**Cell synchronization**. T98G and HME cells were seeded at a low density supplemented with serum-containing MEGM or DMEM medium. The next day, cells were washed with 1× PBS and maintained in serum-deprived medium for 3 days. On the third day, cells were released into the cell cycle using a fresh serum-containing medium and subsequently collected every few hours.

**siRNAs**. All siRNAs were synthesized in Dharmacon Inc and Qiagen: siCtrl 5′-GCGCGCUUUGUAGGAUUCG-3′, siBRCA1: 5′- AGAUAGUUCUACCAGUA AAUU-3′, and siBRCA1 3′-UTR (Qiagen, SI02664368) 5′-CAUACAGCUUCAU AAUAAUU-3′. siRNA oligos were introduced at a final concentration of 20 nM by reverse transfection using RNAiMax (Invitrogen), according to the manufacturer's instructions. Cells were then incubated for 72 h following transfection.

**Plasmid vectors**. Transfection of all vectors was performed using Lipofectamine 2000 (Invitrogen), according to the manufacturer's instructions. The GFP-RNAse H1 vector was described[30]. The V5-RNAse H1 (111906) vector was purchased from Addgene. GST-tagged, truncated BRCA1 fragments (F3 region) were PCR amplified and cloned into the GST bacterial expression vector, pGEX-5X-3 (GE Healthcare).

The BRCA1-ΔNLS1 and BRCA1-R506S (HA-, and GST-tagged) mutants were generated with the QuickChange II Site-directed Mutagenesis Kit (Agilent) using the following primers.

ΔNLS1 Forward: 5′-CCTCACAAATAAATTACCTACATCAGGCCTTC-3′, ΔNLS1 reverse: 5′-GAAGGCCTGATGTAGGTAATTTATTTGTGAGG-3′.

R506S forward: 5′-CAAATAAATTAAAGCGTAAAAGTAGACCTACATCA GGCCTTCATC-3′, R506S reverse: 5′-GATGAAGGCCTGATGTAGGTCTACTT TTACGCTTTAATTTATTTG-3′.

**CRISPR**. The lentiCRISPR-v2 vector (Addgene no. 52961) was used for cloning of the *BRCA1* gRNA (gBRCA1: GAACTCTGAGGACAAAGCAG, exon 7). HME cells were transfected with the gRNA-containing vector in a six-well plate (1 μg/well) using Lipofectamine 2000 and a respective medium. Two days later, cells were split into 10 cm plates at a various density. Following an overnight incubation, fresh, puromycin-containing medium was added (1.5 μg/ml). Four days later, medium without puromycin was added, and the cells were cultivated for 10–14 days. Colonies were then picked and expanded.

Characterization of the CRISPR clones was assessed by PCR amplification of the targeted region and cloning of the PCR-amplified fragments into pCR™ 2.1-TOPO® TA vector (Invitrogen). Genomic alterations in the targeted region were revealed by Sanger sequencing.

**Immunoblotting**. Cells were lysed for 45 min on ice in NETN-400 lysis buffer (400 mM NaCl, 20 mM Tris-HCl pH 7.5, 0.5 mM EDTA, 0.5% NP40, and 10% glycerol) with proteinase and phosphatase inhibitors (Thermo Scientific) followed by centrifugation. A total of 30–50 μg of cell extract were used, and immunoblot was performed according to standard protocols.

**Cellular protein fractionation and Immunoprecipitation**. To isolate and enrich proteins from the chromatin compartment, cells were processed as described[62]. Immunoprecipitations were performed using 120–200 μg of chromatin fractions, diluted three times in IP buffer (100 mM HEPES pH 7.5–8.0, 10 mM NaCl, 1 mM DTT, and protease and phosphatase inhibitors), containing respective antibodies at 4 °C overnight. A total of 30 μl of Protein G magnetic beads (Invitrogen) were then added to IP mixtures for 2 h, washed three times in washing buffer (10 mM HEPES pH 7.5–8.0, 100 mM NaCl, 0.2% Tween-20, 10% glycerol, and protease and phosphatase inhibitors), and analyzed by immunoblotting. For examination of R-loop-dependent protein–protein associations, cells were transfected with GFP-RNAse H1-encoding vector 24 h prior to collection of cells for IP analyses.

The antibodies used in pulldowns and immunoblotting are listed in Supplementary Table 2.

Uncropped gels are provided in the Source data file.

**GST-BRCA1 fragment binding experiments**. GST-BRCA1 fusion proteins were synthesized in *Escherichia coli* and purified on glutathione–sepharose beads (GE healthcare). NETN 250 (250 mM NaCl) precleared HeLa whole-cell extracts (XRN2 binding) or HeLa chromatin fractions (TERRA binding) were incubated with 2 μg of bead-bound GST fusion proteins or GST beads only. The immune complexes were then analyzed by SDS gel electrophoresis and immunoblotting.

Truncated GST-BRCA1-F3 fusion proteins or BRCA1 GST-F2-ΔNLS1 and GST-F2-R506S fusion proteins were generated in BL21(DE3) *E. coli* (Life Technologies) for 16 h at 30 °C using the Overnight Express Autoinduction system (Novagen). Bacterial pellets were incubated in BugBuster Master Mix (Millipore) supplemented with 1 KU/ml rLysozyme (Millipore), benzonase 25 U/ml (Millipore), and protease inhibitors for 30 min at room temperature (RT) at constant rotation and subsequently cleared by centrifugation. GST-tagged proteins were purified from bacterial lysates using Novagen BugBuster GSTBind Purification kit (Millipore) and used for TERRA pulldown assays.

**TERRA RNA pulldown assay**. RNA affinity-based assay was performed as described[62]. Briefly, 20 pmol of biotinylated RNA oligonucleotides, (UUAGGG)$_8$ and (CCCUAA)$_8$, were incubated with 200 μg of HeLa chromatin fractions for 4 h at 4 °C with gentle rotation and subsequently washed three times with washing buffer (20 mM Tris-HCl pH 7.5, 10 mM NaCl, 0.1% Tween-20, 1 U/μl of RNA-seOUT, and protease and phosphatase inhibitors). RNA–protein complexes were then analyzed by SDS gel electrophoresis and immunoblotting.

**DNA–protein affinity assay**. DNA–protein-binding assays were performed using biotinylated (CCCTAA)$_8$, (TTAGGG)$_8$, or (TTTGCG)$_8$ DNA probes (20 pmol), bound to streptavidin-coated magnetic beads in binding buffer (100 mM NaCl and 10 mM Tris-HCl pH8.0) with continuous shaking for 30 min at RT. Then, probe-containing beads were incubated with 100 μg of HeLa chromatin fractions, diluted three times with protein-binding buffer (10 mM Tris-HCl pH 7.5, 10 mM NaCl, 10 μg/ml BSA, 10% NP40, and protease and phosphatase inhibitors), for 30 min at RT. The protein–DNA complexes were then retrieved with a magnet, washed three times with protein-binding buffer and analyzed by immunoblot.

**TERRA hybrid pulldown assay**. To generate TERRA hybrids, 10 μM of each oligonucleotide (TERRA RNA (biotinylated) (UUAGGG)$_8$ and C-rich DNA (CCCTAA)$_8$) were mixed and heated for 5 min at 95 °C, and cooled down gradually to RT. Hybrids were pre-treated with mock- or RNAse H1 enzyme (NEB) for 1 h at 37 °C prior to the assay. The following steps were performed as described above for the RNA pulldown assay.

**Combined immunofluorescence-RNA or DNA FISH**. IF assays were performed according to standard protocols. Primary antibodies used for IF analyses are listed in Supplementary Table 2. Following incubation with a secondary antibody, cells were washed with PBS and subsequently fixed with 4% PFA–PBS for 10 min at RT. For RNA staining, cells were rinsed briefly with 1× PBS and then incubated with hybridization mix (1:500 Tel-C-(CCCTAA)-FAM or Tel-C-(CCCTAA)-Cy3 probe, 50% formamide, 2× SSC, 2 mg/ml BSA, 10% dextran sulfate, 1 mg/ml yeast tRNA, and 10 mM vanadyl ribonucleoside complex) for 18 h in a humidified chamber at 39 °C. Cells were washed with 2× SSC-50% formamide three times at 39 °C for 5 min each, three times in 2× SSC at 39 °C for 5 min each, and once in 2× SSC and once in 4× SSC at RT for 10 min each.

For DNA staining, following a second fixation in 4% PFA–PBS, cells were rinsed briefly with 1× PBS and sequentially dehydrated in 70, 90, and 100% ethanol for 5 min each. Cells were then incubated in hybridization mix (1:500 Tel-C-FAM probe, 70% formamide, 10 mM Tris-HCl pH 7.2, and 0.5% Roche blocking reagent) at 80 °C for 5 min followed by a 2-h incubation at RT. Cells were washed two times with 70% formamide, 10 mM Tris-HCl pH 7.2 at RT for 15 min each, followed by three washes with 1× PBS and mounting with Prolong gold DAPI (Invitrogen). Images were captured with a Yokogawa spinning disk confocal on a Nikon Eclipse-TI inverted microscope and processed with ImageJ software.

**RNA extraction**. RNA (total or from RIP beads) was isolated using TRIzol reagent (Life Technologies), according to the manufacturer's instructions. RNA was subsequently treated twice with DNAse using a TURBO DNA-free kit (Life Technologies).

**Slot blot**. A total of 1 µg of U2OS RNA and 5 µg of HME or HELA RNA were denatured in 50% formamide by heating at 65 °C for 15 min, incubated for 2 min on ice, and spotted onto a positively charged nylon membrane. Membranes were UV-crosslinked (120 mJ) and hybridized overnight in ULTRAhyb hybridization buffer (Invitrogen) with DIG-labeled TERRA (CCCTAA)₅ probe at 42 °C. Membranes were washed and blocked using a DIG wash and block buffer set (Roche). The hybridization signal was revealed using anti-DIG-alkaline phosphatase antibody (Roche) and CDP-Star (Roche). DIG-labeled 18 S rRNA probe (5′-CCATC CAATCGGTAGTAGCG-3′) was used for normalization.

DNA samples (ChIP/DRIP experiments) were denatured at 95 °C for 10 min, incubated on ice for 2 mins, and slot blotted onto nylon membranes. Hybridization and detection steps were performed as described above. Telomere DNA was detected with either DIG-(TTAGGG)₅ probe (for the C-rich strand) or (CCCTAA)₅ probe (for the G-rich strand). An Alu probe (5′-GTGATCCGC CCGCCTCGGCCTCCCAAAGTG-3′) was used for ChIP experiments.

**Southern blot**. Telomere length (TRF) analysis was performed using genomic DNA digested with HinfI and RsaI. TERRA promoter methylation analysis was performed using genomic DNA digested with either MspI or HpaII. Digested DNA was electrophoresed in 0.8% (for TRF analysis) or 1.5% (for methylation analysis) agarose gels. For both analyses, DNA was transferred to nylon membranes and hybridized at 42 °C overnight using DIG-(CCCTAA)₅ probe for telomeric sequences or DIG-labeled TERRA promoter specific probe, generated using a PCR DIG Probe synthesis kit (Roche). Membranes were washed, blocked, and developed, using DIG wash and block buffer set (Roche), as described above.

**γH2AX staining at chromosome ends**. γH2AX staining of metaphase spreads was carried out as in[10]. Briefly, prior to staining, cells were incubated with 0.05 µg/ml colcemid (KaryoMAX 15212012) for 4 h. Cells were then centrifuged for 5 min at $500 \times g$ and the cell pellet was resuspended in 0.075 M KCl at a concentration of $5 \times 10^4$ cells ml$^{-1}$. Cells were incubated in the hypotonic solution at 37 °C for 20 min exactly and cytocentrifuged for 10 min at 2000 RPM with medium acceleration on a Cytospin 3 centrifuge (SHANDON). Following fixation for 10 min in 4% PFA–PBS, slides were rinsed in water and permeabilized with KCM buffer (120 mM KCl, 20 mM NaCl, 10 mM Tris pH 7.5, and 0.5% Triton) for 30 min at RT. The slides were blocked with 3% BSA–0.2% Triton-PBS for 30 min at 37 °C. Slides were then incubated for 1 h at RT with γH2AX antibody in blocking solution. Slides were then washed three times for 5 min with 1× PBST and incubated with a secondary antibody for 1 h at RT. After three washes in 1× PBST, the slides were fixed in 4% PFA–PBS, and telomere DNA FISH staining was performed as described above.

**DNA FISH staining in metaphase spreads**. Cells were treated with colcemid (0.4 µg/ml) for 4–6 h after which cells were harvested and incubated in 0.075 M KCl at 37 °C for 20 min. Chromosomes were fixed in cold methanol/acetic acid (3:1), spread on glass slides, and air-dried overnight. The slides were rehydrated, fixed with 4% PFA–PBS, and treated with 200 µg/ml RNAse A for 30 min at 37 °C. Slides were then digested with pepsin (1 mg/ml) for 10 min at 37 °C following fixation with 4% PFA–PBS. Telomere DNA staining was carried out as described above.

**Chromatin immunoprecipitation**. ChIP steps were performed as previously described[22], using 15–20 µg of sheared chromatin DNA incubated overnight at 4 °C with respective antibodies (Supplementary Table 2). DNA was purified with a PCR purification kit (Qiagen). ChIP samples were then subjected to a slot blot for analysis of telomere association (as described above) or qRT-PCR (see Supplementary Table 3) for examination of subtelomeric regions.

**Quantitative RT-PCR**. A total of 3 µg of RNA were used to generate cDNA using SuperScript IV VILO Master Mix (Invitrogen) and RT-specific primers (Supplementary Table 3). Quantitative RT-PCR was performed in QuantStudio 6 Flex System (Applied Biosystems), using PowerUP SYBR Green Master Mix (Applied Biosystems) and the primers of interest. A cDNA standard curve was used in order to generate data with the assistance of Applied Biosystems QuantStudio software.

**DRIP**. U2OS cells were subjected to DRIP experiments without a crosslinking step, as described in[30]. Genomic DNA was pretreated with mock/RNAse H1 (NEB) for 2 h at 37 °C and sonicated for 30 s on/off for five cycles, using the Bioruptor Pico (Diagenode). A total of 10–15 µg of DNA were used for IP with 2.5 µg of S9.6 antibody. The IPd DNA were then subjected to a slot blot analysis using DIG-(TTAGGG)₅ probe or qRT-PCR (see Supplementary Table 3).

**RNA immunoprecipitation**. Chromatin fractions used for RIP were prepared as described above in the presence of RNAseOUT in all buffers. Relevant extracts were incubated with the BRCA1 primary antibody (SD118) for 3 h at 4 °C, followed by incubation with Protein G Dynabeads (Invitrogen) for 2 h at 4 °C with subsequent washes using washing buffer (10 mM HEPES pH 7.5–8.0, 100 mM NaCl, 0.2% Tween-20, and 10% glycerol). TRIzol reagent was added directly to the beads, and RNA extraction was carried out as previously described.

**Protein pulldown and nanoLC–MS analysis**. HME cells with inducible shScramble or shBRCA1 were grown for two weeks in doxycycline-containing medium to induce BRCA1 depletion. Chromatin fractions were obtained using Subcellular protein fractionation kit (Thermo Scientific), and used for pulldowns of endogenous BRCA1 and TRF2 proteins together with IgG control. Proteins were digested on-beads with trypsin after reduction (10 mM DTT) and alkylation (22.5 mM IAA). After RP/SCX cleanup, peptides were subjected to nanoLC–MS[63], using a NanoAcquity Sample Manager and UPLC pump (Waters, Milford, MA) interfaced with a QExactive mass spectrometer (ThermoFisher Scientific, San Jose, CA). The mass spectrometer was programmed to perform MS/MS (resolution = 15 k, max fill time = 100 ms, target = 1E5, collision energy = 27%, isolation window = 1 Da) on the ten most abundant ions in each MS scan, as well as targeted MS/MS on three proteotypic BRCA1 peptides (resolution = 15 k, max fill time=100 ms, target = 2E5, collision energy = 27%, isolation window = 1.8 Da). Raw data files were converted to.mgf using multiplierz scripts[64], and searched using Mascot version 2.6.1 against a forward reverse human protein database (Uniprot). Search parameters specified fixed carbamidomethylation of cysteine, variable oxidation of methionine, trypsin specificity, a precursor ion tolerance of 10 p.p.m., and a product ion tolerance of 25 m.m.u. Results were filtered to a 1% FDR, and background proteins from the IgG analyses were subtracted from each of the TRF2/BRCA1 IPs. Remaining peptides were mapped to genes using the Pep2Gene utility[64]. A list of candidate BRCA1/TRF2-associated proteins was obtained by filtering for genes mapped by two or more unique peptides.

Venn diagram (http://bioinformatics.psb.ugent.be/webtools/Venn/) was used to obtain a list of overlapping proteins in TRF2 and BRCA1 IPs. Gene Ontology analysis of the shared proteins from the TRF2/BRCA1 IPs was performed using PANTHER (v.14.).

**Statistics and reproducibility**. All experiments were independently repeated at least three times (except Supplementary Fig. 5e, f). Student's $t$ test (two-tailed) was used to calculate $p$ values. Data were reported as mean ± SD. Prism 8 software was used to generate graphs.

**Reporting summary**. Further information on research design is available in the Nature Research Reporting Summary linked to this article.

## Data availability

All relevant data are available from the authors upon reasonable request. Raw mass spectrometry data (Supplementary Table 1) files are available for download at ftp:// massive.ucsd.edu/MSV000087276/. Uniprot database https://www.uniprot.org was used for identification of proteins using Mass spectrometry data. MassIVE database https:// www.massive.ucsd.edu was used for upload of Raw mass spectrometry data. Source data are provided with this paper.

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

## Acknowledgements

We wish to thank Dr. Shailja Pathania (UMass Boston) for providing HME cells. We thank all members of the Livingston group and Dr. David Pellman (Dana-Farber Cancer Institute) for their very helpful advice. We are also grateful to Dr. Joan Brugge and members of her group and to Drs. Myles Brown, Kornelia Polyak, and Robert Weinberg for their valuable suggestions. This work was supported by grants from the National Cancer Institute/NIH-Mechanisms of Breast Development and Carcinogenesis (P01CA080111), BRCA1 Function in Post Damage Foci (R01CA136512), Deciphering the Mechanism Underlying BRCA1 Breast Cancer Development (5R35CA242143-02), the Gray Foundation, the Breast Cancer Research Foundation (BCRF-19-101), The Susan G Komen Foundation for the Cure (SAC140022), the BRCA Foundation, Evan M. and Cynthia Goldberg, and the Murray Winston Foundation.

## Author contributions

J.V. and D.M.L. conceived the project with help from E.H. J.V. designed and performed most of the experiments. L.J.G. and F.O.A. assisted with qRT-PCR experiments. B.L. generated and generously shared CRISPR-modified HME BRCA1 knockout cells and performed GST-binding (XRN2) experiments. Q.K. performed cell cycle FACS analyses and helped to design plasmid vectors. V.V.B. assisted with HR FACS analyses. M.H. and Z.L. assisted with cell culture analyses. A.P.C. helped to design plasmid vectors and shared reagents. S.B.F. and J.A.M. performed nanoLC–MS analysis. E.H. generously shared pull-down protocols. J.V. and D.M.L. wrote the manuscript. All authors read the manuscript.

## Competing interests

D.M.L. is a Science Partner of and a Consultant to Nextech Invest (Zurich Switzerland); a consultant to Constellation Pharma, Boston, MA; a consultant to the Novartis Institute of Biomedical Research; a Scientific Committee Member of the Pezcoller Foundation (Trento, IT), and chairman of the Scientific Advisory Board (SAB) of the MIT Cancer Center, and a member of the SAB of The Sidney Kimmel-Johns Hopkins Cancer Center, and a Scientific Advisor to Cancer Research, UK. The remaining authors declare no competing interests.
