## [Peer Review File · Nature Communications]

REVIEWER COMMENTS

Reviewer #1 (Remarks to the Author):

Vohhodina and colleagues report a novel function of BRCA1 in regulating TERRA R-loops and transcription. First, they show that BRCA1, XRN2 and SETX interact with telomeres, and that this interaction is dependent on R-loops. Suppression of BRCA1 induces DNA damage at telomeres. The authors then demonstrate that BRCA1 interacts with TERRA, and that suppression of BRCA1 is associated with increased TERRA transcription and increased TERRA R-loops at telomeres, which results in replication stress. Finally, they identify a short region in BRCA1 that is responsible for its direct binding to TERRA. Mutations within this domain phenocopy the loss of BRCA1 on accumulation of R-loops and replication stress at telomeres.

The results are novel and of broad interest for the readers of Nature Communications. The data presented here are very convincing and the study is truly well done.

The only major flaw of the manuscript is the interpretation that BRCA1 directly regulates TERRA transcription: TERRA transcription is regulated during the cell cycle and through telomere length; longer telomeres express more TERRA (shown in human cells: Arnoult et al. NSMB 2012, and in yeast: Moravec et al. EMBO rep 2016, Graf et al. Cell 2017). It is therefore possible that the increase in TERRA levels is due to either cell cycle perturbation or, more likely, telomere shortening. Indeed, the authors show that BRCA1 suppression leads to increased levels of telomere-free ends. Furthermore, the model proposed here (not in Figure 6 but in the manuscript, including lines 339-341, lines 364-366, and discussion) is that BRCA1 suppression results in elevated TERRA transcription, which in turn leads to R-loop accumulation and replication stress. However, treatment of siBRCA1 cells with RNaseH1 leads to a reduction in TERRA expression (Figure 3a). Therefore, it is possible that, instead, BRCA1 solely regulates R-loops. Suppression of BRCA1 would result in an accumulation of R-loops, leading to replication stress, telomere shortening and consequently increased TERRA expression.

The authors need to address this point, for example by doing (some of) the following experiments:

- Does BRCA1 bind TERRA promoters?
- Do a TERRA qPCR upon siBRCA1 vs siBRCA1+RNaseH1 - if BRCA1 directly regulates TERRA promoter, RNaseH1 should have no effect.
- Show telomere length (by TRF or qPCR) upon siBRCA1.
- Look at TERRA expression upon XRN2 or SETX knockdown, as these would be expected to help BRCA1 remove R-loops but not regulate TERRA promoters
- Quantify TERRA by qPCR upon siBRCA1 vs siBRCA1+BRCA1 Δ NLS – this could serve as a separation of function mutant. If the author's model is correct

Other major points:

Figure 2F: The authors use U2OS cells to test whether TERRA/BRCA1 co-localizations are reduced upon RNaseH1 treatment. Although I understand the advantage of using ALT cells here, since they express more TERRA and have more R-loop, the presence of ALT-associated PML bodies likely biases the results. Like most HR proteins, BRCA1 localizes to APBs, which is likely independent of TERRA (although it was never demonstrated). Overexpression of RNaseH1 could displace TERRA from APBs without really affecting the TERRA-BRCA1 interaction. To separate TERRA-BRCA1 interaction from TERRA-BRCA1 colocalization in APBs, the authors need to perform the same experiment but add PML IF, and count TERRA-BRCA1 co-localization within and, more importantly, outside of APBs.

Figure 4a-d: Same as 2F for 4a and b. The authors need to show that the TERRA-53BP1 and TERRA-pATR colocalizations occur outside of APBs and that the increased number of colocalizations upon siBRCA1 are not simply the result of an increased number of APBs.

Figure 5e-h: The authors mutate the first NLS of BRCA1 and show that these mutants no longer interact with TERRA. Although the second NLS is expected to be sufficient for BRCA1's nuclear localization, that should be formally demonstrated (for example by doing an IF against HA and showing it is indeed in the nucleus).

Other points:

- Lines 253-256 and 501: The authors see two peaks in the TERRA-BRCA1 interaction (Fig2K), in early S and late S/G2, and argue that these likely correspond to when telomeres replicate, citing a 2010 Plos Genet paper. However, this paper shows that, while some telomeres replicate early in S and others in late S/G2, telomeres replicate all over S phase, the majority of them being in mid-S. Alternatively, when analyzing BrdU incorporation at telomeres, Verdun and colleagues (Cell, 2006) have shown that, beyond this bulk replication, there is a second peak in late S/G2, that is believed to correspond to the opening of the T-loop and replication of the very ends of telomeres. Therefore, an alternative explanation of these 2 peaks could be the requirement for BRCA1 to facilitate replication when TERRA is very high, in early S, and in late S/G2 when telomeres unfold the T-loop to terminate replication.

In summary, I believe that this is a beautiful study providing convincing evidence of a novel function for BRCA1 in removing TERRA R-loops, thereby preventing replication stress at telomeres. These findings shed light on an unanticipated function of BRCA1, which could participate in the high genome instability associated with BRCA1 mutations. The authors need to perform the experiments required to determine whether BRCA1 also directly regulate TERRA transcription or whether TERRA upregulation upon BRCA1 knockdown is an indirect effect of telomere shortening.

Reviewer #2 (Remarks to the Author):

I have carefully read the manuscript by Vohhodina et al that aimed at studying the roles of the observed interaction between BRCA1 and TERRA telomeric RNA in human cells. Many experiments have been performed that, in my view, are not properly analyzed. I have major problems with the interpretation of the observations and the Abstract itself is not supported by the data. A simple explanation for the observations made by the authors may be that, upon BRCA1 depletion (or expression of BRCA1 mutants), telomere free ends increase (as shown previously and confirmed here). This, in turn, may result in increased transcription from the shortened telomeres, and increased formation of R-loops, in line with the previous reports that increased telomeric transcription is associated with more R-loops (see for instance results in ICF cells). Having more R-loops may, in turn, further promote telomere dysfunction and increase gH2AX foci. In this model, BRCA1 may therefore not play any role in directly repressing or repairing R-loops. The Abstract states that BRCA1 promotes the repair of R-loop-associated damage together with XRN2, but there is absolutely no demonstration of this and data are purely correlative. Interpretation is only based on the interaction between BRCA1 and XRN2 that is lost upon RNaseH1 expression. That XRN2 interaction with telomeres is reduced upon RNase H1 expression is not that surprising since R-loops provide binding substrates for XRN2. In the proposed model, how do authors explain that BRCA1 inhibits TERRA transcription? In the discussion, authors discuss about the possible impact of BRCA1 deficiency on subtelomeric promoter DNA methylation level. Transient knock-down with siBRCA1 is unlikely to modify DNA methylation levels, but this should be tested by bisulfite experiments. Alternatively, the binding of BRCA1 to subtelomeric promoters should be evaluated.

Figure 1: I am really surprised by the amazing pull-down efficiency obtained against endogenous shelterin proteins without any cross-linking agent. Not only is the pull-down very efficient against the shelterin protein themselves (although some of the antibodies that are mentioned in the Mat & Met section are normally not suitable for IP), but authors also manage to co-IP the endogenous BRCA1 very efficiently despite the fact that i) BRCA1 shows very poor co-localization with telomeres in FISH/IF experiments (Fig S1b) and ii) BRCA1 appears to interact with TRF2 only in late S (Fig 1g). The same conclusions can be drawn for the BRCA1 pull-down experiment that very efficiently recovers all the endogenous shelterin proteins. How do authors explain this apparent discrepancy between co-IP experiments and FISH/IF or cell cycle-regulated interaction between TRF2 and BRCA1? An experiment with synchronized cells should be performed to follow BRCA1 recruitment to the telomeres over the cell cycle (FISH/IF and ChIP) to clarify the results.

Figure 2f-g: the reduced TERRA-BRCA1 foci in RNase H1-overexpressing cells may simply result

from TERRA degradation in these conditions, as shown by the reduced abundance of TERRA foci in panel f and the reduced abundance of TERRA detected by slot blot (Fig S3a). Also, the co-localization of BRCA1 with TERRA foci may simply reflect the co-localization of BRCA1 with telomeres and does not necessarily imply any binding to TERRA. To test whether BRCA1 recruitment to telomeres is affected by R-loops, a Telo-FISH experiment combined with BRCA1 IF should be performed in RNaseH1-overexpressing cells, and not a TERRA-FISH experiment. Of minor note, no control for RNase H1 expression is shown.

Figure 2h-k: I find the TERRA/BRCA1 IP experiments difficult to understand as there is no detectable BRCA1 in the pull-down experiment at 12h, 15h, 23h, 26h or 30h (panel i), while there is TERRA that is detected (panel j). On the other hand, the only time point for which BRCA1 is efficiently pulled-down (18h) does not reveal any TERRA binding.. Importantly, data obtained for T98G cells (Fig. S2e-g) show very distinct results. How do authors explain all this? Why would BRCA1/TERRA interaction be completely lost in the middle of S phase in HME cells? What do we know about BRCA1 recruitment to telomeres in these various stages (see comment above)? Is BRCA1 removed from telomeres in S phase that may possibly explain why the binding to TERRA is lost? But all these data are difficult to reconcile with the observation that RNaseH1-sensitive BRCA1 interaction with TRF2 appears to only happen in late S. This is extremely confusing and not properly analyzed here.

Fig 3c: telomere length should be measured in BRCA1 +/- and -/- cells to evaluate whether the impact on TERRA expression is not linked to distinct telomere lengths in these cells.

Figure 4: does not show that BRCA1 suppresses replication defects at telomeres arising from TERRA R-loops as stated. In fact, the figure only shows that BRCA1 depletion induces telomeric defects, including TFEs and fragile telomeres. Panels a-b and c-d show the same information as TERRA co-localizes with telomeres.

Figure 5: when deleting BRCA1 from its NLS, what happens to the intracellular localization of the protein? If BRCA1 does not enter the nucleus anymore, this results in a situation equivalent to a knock-down.. The same question applies to the R506S mutant. Or this may affect BRCA1 binding to the telomeres. This should be evaluated by looking at the intracellular localization of BRCA1 mutant proteins and at their ability to bind telomeric repeats. I understand that, according to the authors, BRCA1 binding to telomeres may require R-loops, but the pull-down experiment of Fig 2b shows BRCA1 binding to the (CCCTAA)₈ biotinylated oligonucleotide that unlikely forms R-loops.

Reviewer #3 (Remarks to the Author):

Comments on Vohhodina J et al:

In their manuscript (ms), Vohhhodina et al first discovered a robust interaction between BRCA1 and the Shelterin complex. They further identified XRN2 as an important binding partner for both BRCA1 and TRF2. Most importantly, they found that these interactions are largely depend on the presence of R-loops. They then showed that depletion of BRCA1 results in the upregulation of TERRA. Most intriguingly, they demonstrated that BRCA1 binds directly to TERRA via the NLS1. Suppressing the expression of TERRA by BRCA1 seems to inhibit the replication stress at telomeres. Overall, this is a well-written ms and the authors have made some very important discoveries. I recommend for its acceptance if the authors could successfully address the following concerns:

Major points;

For Fig 1:

1. In Fig 1a, the input bands for TRF1 and RAP1 look strange. Please also indicate the percentage of input for all the IPs.
2. In Fig 1d and 1d and other figures, why most of the telomere foci are so much smaller than the γ -H2AX foci
3. In Fig 1k, please show the expression of RNaseH1

4. The telomere status in BRCA1 deficient cells (shRNA and CRISPR) needs to be thoroughly characterized

For Fig 2:

1. The pull-down experiments should also include a DNA-RNA hybrid, which likely is the real substrate for BRCA1
2. In Fig 2g, the plot includes cells with more than 3 foci, but the images shown in Fig 2f – panel HME+mock contain only two colocalised foci. Please replace them.

For Fig 3:

1. Fig 3a needs statistical analysis
2. For the qPCR analysis of TERRA expression at different chromosomes in three different cell lines, there is not one site that is consistently upregulated across all three cell lines when BRCA1 is depleted. Please explain these surprising results.

For Fig 4:

1. 53BP1 foci is not the best marker for replication stress. Please replace it with phospho-(S317 or S345)-Chk1
2. For Figs 4c and 4d, the authors need to show that by overexpression RNaseH1, the pATR foci are reduced, which would then indicate that it is the R-loops that cause the replication stress

For Fig 5:

1. The authors need to demonstrate whether the NLS1 of BRCA1 is important to suppress the replication stress at telomere

REVIEWERS' COMMENTS

Reviewer #1 (Remarks to the Author):

Vohhodina and colleagues report a novel function of BRCA1 in regulating TERRA R-loops and transcription. First, they show that BRCA1, XRN2 and SETX interact with telomeres, and that this interaction is dependent on R-loops. Suppression of BRCA1 induces DNA damage at telomeres. The authors then demonstrate that BRCA1 interacts with TERRA, and that suppression of BRCA1 is associated with increased TERRA transcription and increased TERRA R-loops at telomeres, which results in replication stress. Finally, they identify a short region in BRCA1 that is responsible for its direct binding to TERRA. Mutations within this domain phenocopy the loss of BRCA1 on accumulation of R-loops and replication stress at telomeres.

The results are novel and of broad interest for the readers of Nature Communications. The data presented here are very convincing and the study is truly well done.

We would like to thank the reviewer for valuable suggestions, which helped to improve the flaw of the manuscript and to strengthen the data.

The only major flaw of the manuscript is the interpretation that BRCA1 directly regulates TERRA transcription: TERRA transcription is regulated during the cell cycle and through telomere length; longer telomeres express more TERRA (shown in human cells: Arnoult et al. NSMB 2012, and in yeast: Moravec et al. EMBO rep 2016, Graf et al. Cell 2017). It is therefore possible that the increase in TERRA levels is due to either cell cycle perturbation or, more likely, telomere shortening. Indeed, the authors show that BRCA1 suppression leads to increased levels of telomere-free ends. Furthermore, the model proposed here (not in Figure 6 but in the manuscript, including lines 339-341, lines 364-366, and discussion) is that BRCA1 suppression results in elevated TERRA transcription, which in turn leads to R-loop accumulation and replication stress. However, treatment of siBRCA1 cells with RNaseH1 leads to a reduction in TERRA expression (Figure 3a). Therefore, it is possible that, instead, BRCA1 solely regulates R-loops. Suppression of BRCA1 would result in an accumulation of R-loops, leading to replication stress, telomere shortening and consequently increased TERRA expression.

The authors need to address this point, for example by doing (some of) the following experiments:

- Does BRCA1 bind TERRA promoters?

As suggested by the reviewer, we performed:

a) ChIP experiments using anti-BRCA1 antibody to test whether BRCA1 binds to TERRA promoters (**Figure 1f-g**). Indeed, we observed that BRCA1 associates with distal and proximal conserved repeats of TelBam3.4 and TelSau2.0 DNA sequences, which are found at multiple chromosome ends and represent CpG-island TERRA promoters¹ (**Figure 1f-g**).

b) In addition, we also detected a cell cycle-dependent BRCA1 recruitment to CpG-rich TERRA promoters in synchronized T98G cells, in a pattern similar to that of TERRA expression (**Figure 2k-l**), suggesting that BRCA1 participates in regulation of TERRA transcription/R-loops.

c) Moreover, we observed that BRCA1 binding to CpG-rich TERRA promoters and telomeric regions is mainly driven by R-loop formation, since exposure to RNase H1 decreased BRCA1

abundance at these regions (**Figure 1f-g**). Consequently, depletion of BRCA1 led to increased levels of R-loops across promoter, subtelomeric and telomeric regions (**Figure 1h**). All together, these data suggest an R-loop-, and a cell cycle-dependent BRCA1 binding to TERRA promoters.

We previously suggested that BRCA1 regulates TERRA transcription based on elevated abundance of TERRA and increased RNA PolIII binding to subtelomeric and telomeric regions following BRCA1 depletion (**Figure 3a-f**). Our new sets of data in support of the aforementioned observations regarding increased TERRA transcription in BRCA1-deficient cells further reveal that:

- a) BRCA1 depletion leads to a change in histone-mark distribution at promoter and telomeric regions (reduced H3K9me3 and H3K27me3, and increased H3K4me3 and H4K16ac occupancies), making the chromatin more accessible and favorable for RNA PolIII binding (**Figure 3g-i**).
- b) BRCA1 depletion results in hypomethylation of CpG-island TERRA promoters (confirmed by methylation-specific Southern blot), which is likely to be a result of diminished recruitment of DNMT3b methyltransferase to these regions (**Figure 3j-k, Supplementary Figure 4l-n**).
- c) Increased TERRA transcription is unlikely to be caused by cell cycle perturbation in the absence of BRCA1 (**Supplementary Fig. 4k**)

Interestingly, in CpG-island-containing promoters, R-loop formation negatively correlates with DNA methylation and thus promotes transcription activation². R-loop structure at CpG promoters is believed to act as a binding site for factors involved in transcription activation (such as H3K4me3, H4K20me1, H3K36me3, H3K79me2, RNA PolII) or as a suppressor of factors associated with transcriptional silencing and methylation (such as repressive histone mark H3K27me3 and DNMTs)^{3,4}.

Since we observed R-loop-dependent binding of BRCA1 to CpG-rich TERRA promoters (**Figure 1g**) as well as increased R-loop-dependent expression of TERRA in the absence of BRCA1 (**Figure 3a-b, Supplementary Fig. 4d-e**), we suggest that R-loops at TERRA promoters (and at telomeric regions) are the prime substrates for BRCA1 and serve as important regulatory elements of TERRA expression. Consequently, abundance of TERRA R-loops in the absence of BRCA1 leads to increased TERRA transcription and consequent DNA damage:

+ **BRCA1** → ↓ R-loops at CpG promoters → ↓ TERRA transcription → ↓ DNA damage at telomeres
- **BRCA1** → ↑ R-loops at CpG promoters → ↑ TERRA transcription → ↑ DNA damage at telomeres

- Do a TERRA qPCR upon siBRCA1 vs siBRCA1+RNaseH1 - if BRCA1 directly regulates TERRA promoter, RNaseH1 should have no effect.

We performed TERRA qRT-PCR in mock-, and RNase H1-treated control and BRCA1-depleted cells, which can be found in **Supplementary Fig. 4d-e**. We observed that upon treatment with RNase H1 TERRA expression in BRCA1-deficient cells decreased to a level, similar to that of control cells (except the chromosome 7p). Therefore, as we suggested above, we assume that BRCA1 regulates R-loops at TERRA CpG-island promoters, which are likely to be the drivers of TERRA transcription¹. Depletion of R-loops with RNase H1, therefore,

reduces TERRA abundance in BRCA1-depleted cells to a ‘baseline’ level on almost all the chromosomes tested (**Supplementary Fig. 4d-e**).

- Show telomere length (by TRF or qPCR) upon siBRCA1.

As suggested by the reviewer, we performed a telomere length analysis by Southern blot in four, different cell lines (**Figure 3l-m, Supplementary Figure 4o**). We observed a significant reduction of telomere length in transiently BRCA1-depleted U2OS and CRISPR-mutated BRCA1^{+/-} and BRCA1^{-/-} cells. Since these BRCA1-deficient cells exhibited significantly upregulated TERRA expression (**Figure 3c-d**), we suggest that elevated TERRA transcription (partially driven by R-loops at CpG-island promoters) in the absence of BRCA1 leads to increased replication stress at telomeres (**Figure 5**) and consequent loss of telomere length. Telomere shortening in turn exacerbates TERRA expression⁵⁻⁷.

Shorter telomere length (and higher TERRA expression) in BRCA1^{-/-} cells compared to BRCA1^{+/-} cells could be explained by higher levels of R-loops at TERRA promoters in BRCA1^{-/-} cells (**Figure 3n**) followed by overly increased TERRA expression (transcription) (**Figure 3d**), resulting in shortened telomere length and consequently aggravated TERRA abundance.

BRCA1^{+/-} → ↑ R-loops at CpG promoters → ↑ TERRA transcription → ↑ DNA damage at telomeres → telomere shortening → ↑↑ TERRA transcription etc.

BRCA1^{-/-} → ↑↑ R-loops at promoters → ↑↑ TERRA transcription → ↑↑ DNA damage at telomeres → ↑ telomere shortening → ↑↑↑ TERRA transcription etc.

- Look at TERRA expression upon XRN2 or SETX knockdown, as these would be expected to help BRCA1 remove R-loops but not regulate TERRA promoters

We examined TERRA expression in control, and BRCA1-, SETX-, and XRN2-depleted cells by qRT-PCR and by slot blot (**Supplementary Fig. 4f-i**). We observed increased TERRA abundance on various chromosomes in the absence of SETX and XRN2, in a pattern similar to that of BRCA1 depletion. In addition, TERRA expression on certain chromosomes, which was either downregulated or unchanged in the absence of BRCA1, also showed a similar trend in the absence of SETX and XRN2.

Thus, we suggest that R-loop-mediated regulation of TERRA transcription by BRCA1 (and of R-loop-associated DNA damage repair) occurs, perhaps, in cooperation with SETX and XRN2.

- Quantify TERRA by qPCR upon siBRCA1 vs siBRCA1+BRCA1ΔNLS – this could serve as a separation of function mutant. If the author’s model is correct

As suggested by the reviewer, we examined TERRA expression by qRT-PCR in U2OS cells, depleted of endogenous BRCA1 and further transfected with HA-tagged WT and mutant BRCA1 (**Supplementary Fig. 5f**). Reconstitution with WT-BRCA1 decreased TERRA abundance compared to BRCA1-depleted cells (**Supplementary Fig. 5f**). Mutant BRCA1-expressing cells, ΔNLS1 and R506S, exhibited a diminished TERRA abundance compared to BRCA1-deficient cells, suggesting that these BRCA1 mutants are able to suppress TERRA abundance, perhaps at the level of transcription.

In line with these results, binding of these BRCA1 mutants to TERRA promoters remained unchanged (**Supplementary Fig. 5g**). Furthermore, we did not observe a significantly

increased R-loop abundance at TERRA promoters in these mutant BRCA1-expressing cells compared to post-depletion of endogenous BRCA1 (**Supplementary Fig. 5e**). This suggests that the BRCA1 NLS1 region, important for binding to the repetitive UUAGGG sequence of TERRA, is not likely to be defective in regulation of R-loop-mediated TERRA transcription.

Other major points:

Figure 2F: The authors use U2OS cells to test whether TERRA/BRCA1 co-localizations are reduced upon RNaseH1 treatment. Although I understand the advantage of using ALT cells here, since they express more TERRA and have more R-loop, the presence of ALT-associated PML bodies likely biases the results. Like most HR proteins, BRCA1 localizes to APBs, which is likely independent of TERRA (although it was never demonstrated). Overexpression of RNaseH1 could displace TERRA from APBs without really affecting the TERRA-BRCA1 interaction. To separate TERRA-BRCA1 interaction from TERRA-BRCA1 colocalization in APBs, the authors need to perform the same experiment but add PML IF, and count TERRA-BRCA1 co-localization within and, more importantly, outside of APBs.

This is a very good suggestion. As advised by the reviewer, we performed co-staining of TERRA with BRCA1 and PML in mock-, and RNase H1-treated U2OS cells (**Supplementary Fig. 3i-j**). Surprisingly, we observed a very intense co-localization of BRCA1 with TERRA within APBs (as defined by triple co-staining) and failed to identify BRCA1-TERRA foci outside of APBs. RNase H1 exposure further reduced the amount of co-localized BRCA1-TERRA foci within APBs, while BRCA1-TERRA co-staining outside of APBs remained almost undetectable.

These data indicate that BRCA1-TERRA interaction occurs solely in APBs. Since BRCA1-TERRA association is mostly R-loop-dependent (**Figure 2e**), we suggest that BRCA1-TERRA interaction in APBs is driven by damaged ALT telomeres and/or ALT DNA synthesis, since both cases are likely to involve TERRA R-loops⁸.

Alternatively, as mentioned by the reviewer, TERRA could act as a sensor for functional BRCA1 to be recruited to APBs, which contain certain telomeric regions anticipating for repair or replication.

Figure 4a-d: Same as 2F for 4a and b. The authors need to show that the TERRA-53BP1 and TERRA-pATR colocalizations occur outside of APBs and that the increased number of colocalizations upon siBRCA1 are not simply the result of an increased number of APBs.

We performed co-staining of TERRA with pChk1^{S345}/pATR^{Thr1989} and PML in mock-, and RNase H1-treated control and BRCA1-depleted U2OS cells (**Supplementary Fig. 6b-f**). As noticed by the reviewer, we did observe an increased number of PML bodies following BRCA1 depletion (**Supplementary Fig. 6f**). For these experiments, we initially adjusted our calculations to a number of TERRA foci in control and BRCA1-depleted cells, since BRCA1 depletion leads to increased TERRA abundance (please find the calculations and adjusted percentage in the Source data-containing file; and the plots in the main (**Figure 5**) and supplementary (**Supplementary Fig. 6**) figures represent adjusted percentages). Amount of PML foci was increased to roughly the same extent as amount of TERRA foci in the absence of BRCA1 (please see the Source data-containing file).

We observed an increased R-loop-dependent TERRA co-localization with pChk1^{S345}/pATR^{Thr1989} within APBs in BRCA1-depleted cells (even following our adjusted calculations) (**Supplementary Fig. 6b-f**). Therefore, we assume that increased replication

stress at telomeres in the absence of BRCA1 is mainly caused by unresolved TERRA R-loops, which promotes the formation of APBs at affected telomeres. Thus, an intense co-localization of TERRA with pChk1^{S345}/pATR^{Thr1989} within, but not outside of, APBs, perhaps, explains this phenomenon (intense TERRA-PML-pATR/pChk1 co-localization).

Figure 5e-h: The authors mutate the first NLS of BRCA1 and show that these mutants no longer interact with TERRA. Although the second NLS is expected to be sufficient for BRCA1's nuclear localization, that should be formally demonstrated (for example by doing an IF against HA and showing it is indeed in the nucleus).

We performed immunostaining of overexpressed HA-tagged WT and mutant BRCA1 using anti-HA antibody, which revealed efficient localization of both BRCA1 mutant vectors to the nucleus despite lack of the NLS1/mutant NLS1 (**Supplementary Figure 5c**).

In addition, we also performed protein fractionation from the U2OS cells, depleted of endogenous BRCA1 and further transfected with HA-tagged WT and mutant BRCA1 (**Supplementary Figure 5d**). Subsequent immunoblot also revealed efficient enrichment of HA-tagged WT and mutant BRCA1 proteins in the nuclear soluble fraction, confirming their sufficient nuclear localization (**Supplementary Figure 5d**). Expression levels of vinculin and HDAC1 were tested to confirm integrity of cytoplasmic and nuclear soluble fractions.

Other points:

- Lines 253-256 and 501: The authors see two peaks in the TERRA-BRCA1 interaction (Fig2K), in early S and late S/G2, and argue that these likely correspond to when telomeres replicate, citing a 2010 Plos Genet paper. However, this paper shows that, while some telomeres replicate early in S and others in late S/G2, telomeres replicate all over S phase, the majority of them being in mid-S. Alternatively, when analyzing BrdU incorporation at telomeres, Verdun and colleagues (Cell, 2006) have shown that, beyond this bulk replication, there is a second peak in late S/G2, that is believed to correspond to the opening of the T-loop and replication of the very ends of telomeres. Therefore, an alternative explanation of these 2 peaks could be the requirement for BRCA1 to facilitate replication when TERRA is very high, in early S, and in late S/G2 when telomeres unfold the T-loop to terminate replication.

We thank the reviewer for a very helpful interpretation, and we updated an explanation of our cell cycle-dependent BRCA1-TERRA interactions in the manuscript text (description of the data from **Figure 2**).

It was recently shown that TERRA can be found at the T-loop junctions, likely adding to their stability⁹, and the role of TRF2 in the formation of T-loops has been described^{10,11}. Since we also observed a specific and robust binding of TRF2 with BRCA1 and XRN2 in late S/G2 phases (**Figure 2j**), we also suggested that the second peak of BRCA1-TERRA interaction in late S/G2 could indicate a formation of functional complex/es that includes BRCA1, XRN2, TRF2, TERRA RNA, and possibly other proteins engaged in unfolding T-loops and/or stabilizing them once telomere replication is complete.

In summary, I believe that this is a beautiful study providing convincing evidence of a novel function for BRCA1 in removing TERRA R-loops, thereby preventing replication stress at telomeres. These findings shed light on an unanticipated function of BRCA1, which could participate in the high genome instability associated with BRCA1 mutations. The authors

need to perform the experiments required to determine whether BRCA1 also directly regulate TERRA transcription or whether TERRA upregulation upon BRCA1 knockdown is an indirect effect of telomere shortening.

We would like to thank the reviewer and we are positive that the newly obtained data will help to better understand BRCA1 functioning at telomeres.

References

1. Nergadze, S.G. et al. CpG-island promoters drive transcription of human telomeres. *RNA* **15**, 2186-94 (2009).
2. Ginno, P.A., Lott, P.L., Christensen, H.C., Korf, I. & Chedin, F. R-loop formation is a distinctive characteristic of unmethylated human CpG island promoters. *Mol Cell* **45**, 814-25 (2012).
3. Ginno, P.A., Lim, Y.W., Lott, P.L., Korf, I. & Chedin, F. GC skew at the 5' and 3' ends of human genes links R-loop formation to epigenetic regulation and transcription termination. *Genome Res* **23**, 1590-600 (2013).
4. Takeshima, H., Yamashita, S., Shimazu, T., Niwa, T. & Ushijima, T. The presence of RNA polymerase II, active or stalled, predicts epigenetic fate of promoter CpG islands. *Genome Res* **19**, 1974-82 (2009).
5. Arnoult, N., Van Beneden, A. & Decottignies, A. Telomere length regulates TERRA levels through increased trimethylation of telomeric H3K9 and HP1alpha. *Nat Struct Mol Biol* **19**, 948-56 (2012).
6. Moravec, M. et al. TERRA promotes telomerase-mediated telomere elongation in *Schizosaccharomyces pombe*. *EMBO Rep* **17**, 999-1012 (2016).
7. Graf, M. et al. Telomere Length Determines TERRA and R-Loop Regulation through the Cell Cycle. *Cell* **170**, 72-85 e14 (2017).
8. Arora, R. et al. RNaseH1 regulates TERRA-telomeric DNA hybrids and telomere maintenance in ALT tumour cells. *Nat Commun* **5**, 5220 (2014).
9. Kar, A., Willcox, S. & Griffith, J.D. Transcription of telomeric DNA leads to high levels of homologous recombination and t-loops. *Nucleic Acids Res* **44**, 9369-9380 (2016).
10. Doksan, Y., Wu, J.Y., de Lange, T. & Zhuang, X. Super-resolution fluorescence imaging of telomeres reveals TRF2-dependent T-loop formation. *Cell* **155**, 345-356 (2013).
11. Sarek, G. et al. CDK phosphorylation of TRF2 controls t-loop dynamics during the cell cycle. *Nature* **575**, 523-527 (2019).

Reviewer #2 (Remarks to the Author):

I have carefully read the manuscript by Vohhodina et al that aimed at studying the roles of the observed interaction between BRCA1 and TERRA telomeric RNA in human cells. Many experiments have been performed that, in my view, are not properly analyzed. I have major problems with the interpretation of the observations and the Abstract itself is not supported by the data.

A simple explanation for the observations made by the authors may be that, upon BRCA1 depletion (or expression of BRCA1 mutants), telomere free ends increase (as shown previously and confirmed here). This, in turn, may result in increased transcription from the shortened telomeres, and increased formation of R-loops, in line with the previous reports that increased telomeric transcription is associated with more R-loops (see for instance results in ICF cells). Having more R-loops may, in turn, further promote telomere dysfunction and increase γ H2AX foci. In this model, BRCA1 may therefore not play any role in directly repressing or repairing R-loops.

The Abstract states that BRCA1 promotes the repair of R-loop-associated damage together with XRN2, but there is absolutely no demonstration of this and data are purely correlative. Interpretation is only based on the interaction between BRCA1 and XRN2 that is lost upon RNaseH1 expression. That XRN2 interaction with telomeres is reduced upon RNase H1 expression is not that surprising since R-loops provide binding substrates for XRN2.

We agree that the statement in our abstract regarding the assisting role of XRN2 in BRCA1-mediated repair of R-loop-associated DNA damage at telomeres was a bit strong. Therefore, we performed a few experiments to further examine importance of XRN2 in resolution of TERRA R-loops in cooperation with BRCA1:

- a)** γ H2AX staining on metaphase spreads in XRN2-depleted cells revealed increased DNA damage at telomeres, a phenotype similar to that of BRCA1 depletion (**Supplementary Fig. 2g-h**).
- b)** We performed qRT-PCR and observed increased TERRA abundance on various chromosomes in the absence of XRN2 as well as of SETX, in a pattern similar to that of BRCA1 depletion (**Supplementary Fig. 4h-i**). In addition, TERRA expression on certain chromosomes, which was either downregulated or unchanged in the absence of BRCA1, also showed a similar trend in the absence of XRN2 and SETX, suggesting that all three proteins could cooperatively regulate abundance of TERRA and its R-loops.
- c)** We observed a BRCA1-mediated association of TRF2 with XRN2 (**Supplementary Fig. 3q**). Since we showed that the aforementioned proteins interact mainly in an R-loop-dependent manner (**Figure 1e**) and depletion of these genes results in an R-loop accumulation (**Supplementary Fig. 1j-k**), we further suggest formation of TERRA R-loop-driven functional complex/es, in which BRCA1 is responsible for mediating interaction of XRN2 with TRF2.

In the proposed model, how do authors explain that BRCA1 inhibits TERRA transcription? In the discussion, authors discuss about the possible impact of BRCA1 deficiency on subtelomeric promoter DNA methylation level. Transient knock-down with siBRCA1 is unlikely to modify DNA methylation levels, but this should be tested by bisulfite experiments. Alternatively, the binding of BRCA1 to subtelomeric promoters should be evaluated.

As suggested by the reviewer, we examined **1)** a possibility of BRCA1 binding to TERRA promoters and **2)** methylation (and chromatin) status of TERRA promoters to understand in detail the mechanism of BRCA1-mediated suppression of TERRA transcription.

1) To test whether BRCA1 binds to TERRA promoters, we performed:

a) ChIP experiments using anti-BRCA1 antibody to test whether BRCA1 binds to TERRA promoters (**Figure 1f-g**). Indeed, we observed that BRCA1 associates with distal and proximal conserved repeats of TelBam3.4 and TelSau2.0 DNA sequences, which are found at multiple chromosome ends and represent CpG-island TERRA promoters ¹(**Figure 1f-g**).

b) In addition, we also detected a cell cycle-dependent BRCA1 recruitment to CpG-rich TERRA promoters in synchronized T98G cells, in a pattern similar to that of TERRA expression (**Figure 2k-l**), suggesting that BRCA1 participates in regulation of TERRA transcription/R-loops.

c) Moreover, we observed that BRCA1 binding to CpG-rich TERRA promoters and telomeric regions is mainly driven by R-loop formation, since exposure to RNase H1 decreased BRCA1 abundance in these regions (**Figure 1f-g**). Consequently, depletion of BRCA1 led to increased levels of R-loops across promoter, subtelomeric and telomeric regions (**Figure 1h**).

All together, these data suggest an R-loop-, and a cell cycle-dependent BRCA1 binding to TERRA promoters.

2) We previously suggested that BRCA1 regulates TERRA transcription based on elevated abundance of TERRA and increased RNA PolII binding to subtelomeric and telomeric regions following BRCA1 depletion (**Figure 3a-f**). Our new sets of data in support of the aforementioned observations regarding increased TERRA transcription in BRCA1-deficient cells further reveal that:

a) CRISPR-modified BRCA1^{+/-} and BRCA1^{-/-} HME cells and transiently BRCA1-depleted U2OS cells exhibited hypomethylation of CpG-island TERRA promoters (confirmed by Southern blot using methylation-specific enzymes *MspI* and *HpaII* and TERRA promoter-specific probe), which is likely to be a result of diminished recruitment of DNMT3b methyltransferase to these regions (**Figure 3j-k, Supplementary Figure 4l-n**).

b) BRCA1 depletion led to a change in histone-mark distribution in promoter and telomeric regions (reduced H3K9me3 and H3K27me3, and increased H3K4me3 and H4K16ac occupancies), making the chromatin more accessible and favorable for RNA PolII binding (**Figure 3g-i**).

Interestingly, in CpG-island-containing promoters, R-loop formation negatively correlates with DNA methylation and thus promotes transcription activation ². R-loop structure at CpG promoters is believed to act as a binding site for factors involved in transcription activation (such as H3K4me3, H4K20me1, H3K36me3, H3K79me2, RNA PolII) or as a suppressor of factors associated with transcriptional silencing and methylation (such as repressive histone mark H3K27me3 and DNMTs)^{3,4}.

Since we observed R-loop-dependent binding of BRCA1 to CpG-rich TERRA promoters (**Figure 1g**) as well as increased R-loop-dependent expression of TERRA in the absence of BRCA1 (**Figure 3a-b, Supplementary Fig. 4d-e**), we suggest that R-loops at TERRA promoters (and at telomeric regions) are the prime substrates for BRCA1 and serve as important regulatory elements of TERRA expression. Consequently, abundance of TERRA R-loops in the absence of BRCA1 leads to increased TERRA transcription and consequent DNA damage:

+ **BRCA1** → ↓ R-loops at CpG promoters → ↓ TERRA transcription → ↓ DNA damage at telomeres
 - **BRCA1** → ↑ R-loops at CpG promoters → ↑ TERRA transcription → ↑ DNA damage at telomeres

Figure 1: I am really surprised by the amazing pull-down efficiency obtained against endogenous shelterin proteins without any cross-linking agent. Not only is the pull-down very efficient against the shelterin protein themselves (although some of the antibodies that are mentioned in the Mat & Met section are normally not suitable for IP), but authors also manage to co-IP the endogenous BRCA1 very efficiently despite the fact that i) BRCA1 shows very poor co-localization with telomeres in FISH/IF experiments (Fig S1b) and ii) BRCA1 appears to interact with TRF2 only in late S (Fig 1g). The same conclusions can be drawn for the BRCA1 pull-down experiment that very efficiently recovers all the endogenous shelterin proteins. How do authors explain this apparent discrepancy between co-IP experiments and FISH/IF or cell cycle-regulated interaction between TRF2 and BRCA1? An experiment with synchronized cells should be performed to follow BRCA1 recruitment to the telomeres over the cell cycle (FISH/IF and ChIP) to clarify the results.

Out of 6 antibodies, anti-POT1 (Abcam, ab124784) and anti-TIN2 (Abcam, ab197894) were not mentioned by the manufacturer as suitable for IP, however we were able to apply these antibodies in our IP analyses. Please find two, representative TIN2 pulldowns using the aforementioned anti-TIN2 antibody below. The IP nr.1 also includes TIN2 pulldown from control and TIN2-depleted cells, exhibiting efficient pulldown of endogenous TIN2 protein from the control sample (please compare IP:TIN2 vs supernatant in siCtrl and siTIN2 sample). Efficiencies of representative TIN2 IPs may vary upon integrity of protein chromatin fractions used for the assays.

Since we performed IP analyses for our studies a few years ago, anti-POT1 (Abcam, ab124784) antibody is no longer produced by the manufacturer. We obtained a new anti-POT1 antibody (Proteintech, 10581-1-AP), and representative IP assays using this antibody, confirming POT1 interaction with BRCA1 and XRN2 as well as its efficient pulldown, can be found below.

IP:POT1 nr.1

IP:POT1 nr.2

Also, below you can find other sets (**a** and **b** (the latter independently performed by Z.L.)) of BRCA1 pulldowns confirming its interaction with XRN2 and shelterin proteins.

Our IF-FISH experiments (BRCA1 co-localization with telomeric DNA) were performed in telomerase immortalized HME cells, derived from a BRCA1^{+/+} tumor-free individual, with a normal karyotype (46 chromosomes). Whereas, IP analyses were mostly performed in HeLa and T98G cells, which exhibit 76-80 and 128-132 chromosomes, respectively, thus likely increasing occurrence of BRCA1-shelterin interactions. In addition, telomeric DNA is coated with numerous molecules of shelterin proteins⁵, thus likely enhancing IP efficiency, whereas BRCA1 localization at telomeric DNA in IF-FISH experiments usually appears as one focus per chromosome. Also, we used different BRCA1 antibodies for IP (SD118 OP107, against

1005-1313aa) and IF (Millipore 07-434, against 1301-1863aa), which again are likely to add to the specificity variation of these methods.

TRF2 pulldown, performed in different phases of the cell cycle in synchronized T98G cells (**Figure 2j**), indeed reveals a solely late S/G2-M interaction of TRF2 with BRCA1 and XRN2. However, it appears that the amount of S/G2 cells in asynchronous population of cells used for IP is enough to observe TRF2-BRCA1 interaction in our ‘unsynchronized’ IPs (please compare ‘uns’ and ‘S/G2-M’ IP bands). It could also explain efficiency of our other IPs (of shelterins and of BRCA1), since they were also performed using unsynchronized cells (**Figure 1a**, **Supplementary Fig. 1a**).

As advised by the reviewer, to further examine BRCA1 recruitment timing to telomeres, we also performed IF-DNA FISH as well as BRCA1 ChIP experiments in synchronized T98G cells. Both assays revealed that BRCA1 localized to telomeres in a cell cycle-dependent manner, increasing in G1/S and becoming maximal in late S-G2/M phases (**Figure 1b**, **Supplementary Fig. 1b-d**). Thus, it appears that while BRCA1 is localized to telomeres during S/G2/M phases, it does not necessarily interact with TRF2 during the whole duration of its presence at these regions (**Figure 2j**). BRCA1 may be involved in interactions with other shelterins or DNA damage/repair-associated proteins during S/G2/M period at telomeres, thus facilitating efficient replication of telomeric DNA.

Figure 2f-g: the reduced TERRA-BRCA1 foci in RNase H1-overexpressing cells may simply result from TERRA degradation in these conditions, as shown by the reduced abundance of TERRA foci in panel f and the reduced abundance of TERRA detected by slot blot (Fig S3a). Also, the co-localization of BRCA1 with TERRA foci may simply reflect the co-localization of BRCA1 with telomeres and does not necessarily imply any binding to TERRA. To test whether BRCA1 recruitment to telomeres is affected by R-loops, a Telo-FISH experiment combined with BRCA1 IF should be performed in RNaseH1-overexpressing cells, and not a TERRA-FISH experiment. Of minor note, no control for RNase H1 expression is shown.

Abundance of TERRA foci has been previously shown to decrease following RNase H1 treatment, as a result of elimination of TERRA-telomeric hybrids⁶. Therefore, we suggested that decreased amount of TERRA-BRCA1 co-localizations following RNase H1 exposure is linked to their R-loop-dependent association, which is likely to be more prevalent in ALT cells compared to telomerase-positive cells ⁶ (**Figure 2f**, **Supplementary Fig. 3f**). Respective immunoblot confirming overexpression of an RNaseH1-encoding vector in U2OS and HME cells for this experiment can be found in **Supplementary Fig. 3e**. R-loop-driven association of BRCA1 and TERRA was also confirmed by the RIP assays (**Figure 2e**, **Supplementary Fig. 3n**).

As suggested by the reviewer, we also performed BRCA1-telomere DNA FISH staining in mock-, and RNase H1-treated U2OS and HME cells (**Supplementary Fig. 3g-h**). We observed the same trend as in our IF-TERRA FISH experiments (**Figure 2f**, **Supplementary Fig. 3f**), thus suggesting that association of BRCA1 with TERRA/telomeric DNA is mainly driven by TERRA R-loops, formation of which is more frequent in ALT cells.

Additionally, we also revealed that TERRA-BRCA1 co-localization occurs solely in APBs (as defined by co-staining with PML bodies) (**Supplementary Fig. 3i-j**). Since BRCA1-TERRA association is mostly R-loop-dependent (**Figure 2e**), we suggest that BRCA1-TERRA interaction in APBs is driven by damaged ALT telomeres and/or ALT DNA synthesis, since both cases are likely to involve TERRA R-loops ⁶.

Figure 2h-k: I find the TERRA/BRCA1 IP experiments difficult to understand as there is no detectable BRCA1 in the pull-down experiment at 12h, 15h, 23h, 26h or 30h (panel i), while there is TERRA that is detected (panel j). On the other hand, the only time point for which BRCA1 is efficiently pulled-down (18h) does not reveal any TERRA binding.. Importantly, data obtained for T98G cells (Fig. S2e-g) show very distinct results. How do authors explain all this? Why would BRCA1/TERRA interaction be completely lost in the middle of S phase in HME cells? What do we know about BRCA1 recruitment to telomeres in these various stages (see comment above)? Is BRCA1 removed from telomeres in S phase that may possibly explain why the binding to TERRA is lost? But all these data are difficult to reconcile with the observation that RNaseH1-sensitive BRCA1 interaction with TRF2 appears to only happen in late S. This is extremely confusing and not properly analyzed here.

We observed a similar trend of decreased BRCA1-TERRA interaction in mid S phase of HME (18h) and T98G (23h) cells (**Figure 2h-i, Supplementary Fig. 3l-m**), potentially due to very low TERRA expression in this particular time frame (**Figure 2l**), since abundant TERRA likely hampers telomeric DNA replication^{7,8}. However, we would not scrupulously compare the obtained RIP results between these cell lines since they are of a different type, as well as the former is a normal cell line, and the latter is cancerous.

BRCA1 binding to telomeric DNA rises in G1/S and occurs throughout S-G2-M phases (**Figure 1b, Supplementary Fig. 1b-d**). BRCA1 association with TERRA also occurs from G1/S up to G2/M, with the two peaks appearing at the G1/S border-early S and late G2/M phases (**Figure 2h-i, Supplementary Fig. 3l-m**). BRCA1-TERRA interaction is further diminished in all phases of the cell cycle upon exposure to RNase H1, with a particular decline in S and G2 (**Supplementary Fig. 3n-o**). Therefore, the two R-loop-dependent peaks of BRCA1-TERRA interaction could be related to BRCA1-mediated facilitation of telomere replication when TERRA expression is high enough, in G1/S-early S and late S/G2 phases. Alternatively, BRCA1-TERRA interaction in late S/G2 phase could be related to the unwinding of the T-loop to successfully complete replication^{9, 10}. Please find the schematic representation of the aforementioned associations below.

It was recently shown that TERRA can be found at the T-loop junctions, likely adding to their stability¹¹, and the role of TRF2 in the formation of T-loops has been described^{12,13}. Since we also observed a specific and robust binding of TRF2 to BRCA1 and XRN2 in late S/G2 phases (**Figure 2j**), we suggested in the manuscript that the second peak of BRCA1-TERRA interaction in late S/G2 could indicate a formation of functional complex/es that includes

BRCA1, XRN2, TRF2, TERRA RNA, and possibly other proteins engaged in unfolding T-loops and/or stabilizing them once telomere replication is complete.

Fig 3c: telomere length should be measured in BRCA1^{+/-} and ^{-/-} cells to evaluate whether the impact on TERRA expression is not linked to distinct telomere lengths in these cells.

We performed a telomere length analysis by Southern blot in our CRISPR-modified BRCA1^{+/-} and BRCA1^{-/-} cells along with BRCA1 depletion in three, other cell lines (**Figure 3l-m, Supplementary Figure 4o**). We observed a significant reduction of telomere length in transiently BRCA1-depleted U2OS and CRISPR-modified BRCA1^{+/-} and BRCA1^{-/-} cells. Since these BRCA1-deficient cells exhibited a significantly upregulated TERRA expression (**Figure 3c-d**), we suggest that elevated TERRA transcription (perhaps partially driven by R-loops at CpG-island promoters) in the absence of BRCA1 leads to increased replication stress at telomeres (**Figure 5**) and consequent loss of telomere length. Telomere shortening in turn exacerbates TERRA expression¹⁴⁻¹⁶.

Shorter telomere length (and higher TERRA expression) in BRCA1^{-/-} cells compared to BRCA1^{+/-} cells could be explained by higher levels of R-loops at TERRA promoters in BRCA1^{-/-} cells (**Figure 3n**) followed by overly increased TERRA levels (**Figure 3d**), resulting in shortened telomere length and consequently aggravated TERRA abundance.

BRCA1^{+/-} → ↑ R-loops at CpG promoters → ↑ TERRA transcription → ↑ DNA damage at telomeres → telomere shortening → ↑↑ TERRA transcription etc.

BRCA1^{-/-} → ↑↑ R-loops at promoters → ↑↑ TERRA transcription → ↑↑ DNA damage at telomeres → ↑ telomere shortening → ↑↑↑ TERRA transcription etc.

Figure 4: does not show that BRCA1 suppresses replication defects at telomeres arising from TERRA R-loops as stated. In fact, the figure only shows that BRCA1 depletion induces telomeric defects, including TFEs and fragile telomeres. Panels a-b and c-d show the same information as TERRA co-localizes with telomeres.

We agree with the reviewer's point and, therefore, we performed co-staining of TERRA with pChk1^{S345}/ pATR^{Thr1989} and TRF2 in mock-, and RNase H1-treated control and BRCA1-depleted U2OS cells, which can be found in **Figure 5a-c** (respective immunoblot confirming expression of RNase H1-encoding vector is shown in **Supplementary Fig. 6a**). An amount of TERRA-TRF2-pChk1^{S345}/ pATR^{Thr1989} co-localized foci fell upon exposure to RNase H1 in BRCA1-depleted cells, suggesting that increased abundance of TERRA R-loops is likely to be the cause of replication stress at telomeres.

In addition, a predominantly R-loop-dependent co-localization of TERRA with pChk1^{S345}/ pATR^{Thr1989} appears to occur solely in APBs (as defined by co-staining with PML bodies) (**Supplementary Fig. 6b-e**). This suggests that increased replication stress at telomeres in the absence of BRCA1 is mainly caused by unresolved TERRA R-loops, which promotes formation of APBs at affected telomeres, as defined by intense co-localization of TERRA with pChk1^{S345}/pATR^{Thr1989} within APBs.

Moreover, we also show that TERRA-interaction associated BRCA1 mutants, ΔNLS1 and R506S, are defective in suppression of replication stress at telomeres, similar to that observed in BRCA1-depleted cells, as defined by increased pChk1^{S345}-TRF2 and pATR^{Thr1989}-TRF2 co-

staining in U2OS cells, depleted of endogenous BRCA1 and further transfected with HA-tagged WT and mutant BRCA1 (**Figure 5d-e, Supplementary Fig. 6g-i**).

Figure 5: when deleting BRCA1 from its NLS, what happens to the intracellular localization of the protein? If BRCA1 does not enter the nucleus anymore, this results in a situation equivalent to a knock-down.. The same question applies to the R506S mutant. Or this may affect BRCA1 binding to the telomeres. This should be evaluated by looking at the intracellular localization of BRCA1 mutant proteins and at their ability to bind telomeric repeats. I understand that, according to the authors, BRCA1 binding to telomeres may require R-loops, but the pull-down experiment of Fig 2b shows BRCA1 binding to the (CCCTAA)₈ biotinylated oligonucleotide that unlikely forms R-loops.

We performed immunostaining of overexpressed HA-tagged WT and HA-mutant BRCA1 using anti-HA antibody, which revealed efficient localization of both BRCA1 mutant vectors to the nucleus despite lack of the NLS1/mutant NLS1 (**Supplementary Figure 5c**).

In addition, we also performed protein fractionation from the U2OS cells, depleted of endogenous BRCA1 and further transfected with HA-tagged WT and mutant BRCA1 (**Supplementary Figure 5d**). Subsequent immunoblot also revealed efficient enrichment of HA-tagged WT and mutant BRCA1 proteins in nuclear soluble fraction, confirming their sufficient nuclear localization (**Supplementary Figure 5d**). Expression levels of vinculin and HDAC1 were tested to confirm integrity of cytoplasmic and nuclear soluble fractions.

Also, we observed increased binding of the BRCA1 mutants to telomeric DNA repeat sequences, perhaps as a result of their retention on telomeric chromatin due to unresolved telomeric R-loops (**Supplementary Fig. 5g-h, Fig. 4g**).

References

1. Nergadze, S.G. et al. CpG-island promoters drive transcription of human telomeres. *RNA* **15**, 2186-94 (2009).
2. Ginno, P.A., Lott, P.L., Christensen, H.C., Korf, I. & Chedin, F. R-loop formation is a distinctive characteristic of unmethylated human CpG island promoters. *Mol Cell* **45**, 814-25 (2012).
3. Ginno, P.A., Lim, Y.W., Lott, P.L., Korf, I. & Chedin, F. GC skew at the 5' and 3' ends of human genes links R-loop formation to epigenetic regulation and transcription termination. *Genome Res* **23**, 1590-600 (2013).
4. Takeshima, H., Yamashita, S., Shimazu, T., Niwa, T. & Ushijima, T. The presence of RNA polymerase II, active or stalled, predicts epigenetic fate of promoter CpG islands. *Genome Res* **19**, 1974-82 (2009).
5. Takai, K.K., Hooper, S., Blackwood, S., Gandhi, R. & de Lange, T. In vivo stoichiometry of shelterin components. *J Biol Chem* **285**, 1457-67 (2010).
6. Arora, R. et al. RNaseH1 regulates TERRA-telomeric DNA hybrids and telomere maintenance in ALT tumour cells. *Nat Commun* **5**, 5220 (2014).

7. Azzalin, C.M., Reichenbach, P., Khoriantuli, L., Giulotto, E. & Lingner, J. Telomeric repeat containing RNA and RNA surveillance factors at mammalian chromosome ends. *Science* **318**, 798-801 (2007).
8. Yehezkel, S., Segev, Y., Viegas-Pequignot, E., Skorecki, K. & Selig, S. Hypomethylation of subtelomeric regions in ICF syndrome is associated with abnormally short telomeres and enhanced transcription from telomeric regions. *Hum Mol Genet* **17**, 2776-89 (2008).
9. Arnoult, N. et al. Replication timing of human telomeres is chromosome arm-specific, influenced by subtelomeric structures and connected to nuclear localization. *PLoS Genet* **6**, e1000920 (2010).
10. Verdun, R.E. & Karlseder, J. The DNA damage machinery and homologous recombination pathway act consecutively to protect human telomeres. *Cell* **127**, 709-20 (2006).
11. Kar, A., Willcox, S. & Griffith, J.D. Transcription of telomeric DNA leads to high levels of homologous recombination and t-loops. *Nucleic Acids Res* **44**, 9369-9380 (2016).
12. Doksani, Y., Wu, J.Y., de Lange, T. & Zhuang, X. Super-resolution fluorescence imaging of telomeres reveals TRF2-dependent T-loop formation. *Cell* **155**, 345-356 (2013).
13. Sarek, G. et al. CDK phosphorylation of TRF2 controls t-loop dynamics during the cell cycle. *Nature* **575**, 523-527 (2019).
14. Arnoult, N., Van Beneden, A. & Decottignies, A. Telomere length regulates TERRA levels through increased trimethylation of telomeric H3K9 and HP1alpha. *Nat Struct Mol Biol* **19**, 948-56 (2012).
15. Moravec, M. et al. TERRA promotes telomerase-mediated telomere elongation in *Schizosaccharomyces pombe*. *EMBO Rep* **17**, 999-1012 (2016).
16. Graf, M. et al. Telomere Length Determines TERRA and R-Loop Regulation through the Cell Cycle. *Cell* **170**, 72-85 e14 (2017).

Reviewer #3 (Remarks to the Author):

Comments on Vohhodina J et al:

In their manuscript (ms), Vohhhodina et al first discovered a robust interaction between BRCA1 and the Shelterin complex. They further identified XRN2 as an important binding partner for both BRCA1 and TRF2. Most importantly, they found that these interactions are largely depend on the presence of R-loops. They then showed that depletion of BRCA1 results in the upregulation of TERRA. Most intriguingly, they demonstrated that BRCA1 binds directly to TERRA via the NLS1. Suppressing the expression of TERRA by BRCA1 seems to inhibit the replication stress at telomeres. Overall, this is a well-written ms and the authors have made some very important discoveries. I recommend for its acceptance if the authors could successfully address the following concerns:

We would like to thank the reviewer for valuable suggestions, which indeed helped to strengthen the data.

Major points;

For Fig 1:

1. In Fig 1a, the input bands for TRF1 and RAP1 look strange. Please also indicate the percentage of input for all the IPs.

We repeated an IP using anti-BRCA1 antibody and obtained more representative immunoblots for RAP1 and TRF1, which can be found in **Figure 1a**. As suggested by the reviewer, we also indicated the percentage of input as well as specified the cellular compartment (chromatin fraction) used in our IP analyses.

2. In Fig 1d and 1d and other figures, why most of the telomere foci are so much smaller than the γ -H2AX foci

This is a very good question. The extent of γ H2AX spreading at telomere-adjacent regions was shown to be \sim 10kb, and opened chromatin was reported to induce spreading of phosphorylated H2AX, resulting in larger γ H2AX foci¹.

We also observed a more accessible chromatin state in BRCA1-depleted cells (as we have shown that BRCA1 regulates TERRA transcription and affects the state of chromatin at subtelomeric and telomeric regions, **Figure 3g-i**). In addition, the average telomere length of the cell lines used in our study varies between 5.5-7kb with a further reduction upon BRCA1 depletion (**Figure 3l-m**). Therefore, we think that this phenomenon (telomere foci are smaller than γ H2AX foci) is related to a wide spreading of γ H2AX combined with a reduction of telomere length in BRCA1-deficient cells. Similarly, an accumulation of large γ H2AX foci at telomeres was observed in senescent human fibroblasts².

In control cells, γ H2AX spread could be caused by random DNA damage at telomeres, which again could lead to chromatin remodeling³. Alternatively, it was reported that heterochromatin does not restrain spreading of γ H2AX even at more distal regions from DNA double-strand breaks¹.

3. In Fig 1k, please show the expression of RNaseH1

Please find the immunoblot confirming overexpression of V5-tagged RNaseH1-encoding vector in HME cells (for γ H2AX staining at telomeres in **Figure 1j-k**) in **Figure 1i**.

4. The telomere status in BRCA1 deficient cells (shRNA and CRISPR) needs to be thoroughly characterized

We performed a Southern Blot to examine telomere length in our CRISPR-modified and inducible shBRCA1-containing HME cells (along with transient BRCA1 depletion in U2OS and HeLa cells) in **Figure 3l-m**. We observed a significant reduction in telomere length in BRCA1^{+/-} and, particularly, in BRCA1^{-/-} cells, while the telomere length was mildly reduced in shBRCA1-depleted HME cells, possibly due to a transient nature of BRCA1 depletion. Although we also performed a transient depletion of BRCA1 in U2OS cells, we observed a significant decrease in telomere length, which is possibly related to defective HR-based elongation of telomeres in this ALT cell line caused by BRCA1 depletion.

For Fig 2:

1. The pull-down experiments should also include a DNA-RNA hybrid, which likely is the real substrate for BRCA1

This is a very good suggestion. We performed TERRA hybrid pulldown assay, which can be found in **Figure 2d**, using the TERRA RNA probe and the C-rich telomeric DNA probe. Indeed, we observed a preferential binding of BRCA1 (and XRN2 as well as robust binding of SETX) to TERRA hybrids, implying that BRCA1 binds TERRA favorably when it forms a DNA-RNA hybrid. RPA34 protein was used as a negative control since it did not exhibit binding to the TERRA probe from **Figure 2c**.

2. In Fig 2g, the plot includes cells with more than 3 foci, but the images shown in Fig 2f – panel HME+mock contain only two colocalised foci. Please replace them.

We thank the reviewer for pointing it out, and the image for HME+mock sample is now replaced with the new one indicating $3 \geq$ TERRA-BRCA1 co-localized foci. Due to space limitations, these IF-FISH images can now be found in **Supplementary Fig. 3f**.

For Fig 3:

1. Fig 3a needs statistical analysis

We added densitometry plot of respective TERRA slot blots and performed statistical analysis, please find it in **Figure 3b**.

2. For the qPCR analysis of TERRA expression at different chromosomes in three different cell lines, there is not one site that is consistently upregulated across all three cell lines when BRCA1 is depleted. Please explain these surprising results.

Variations of TERRA expression among the cell lines and different chromosomes tested in the absence of BRCA1 could be influenced by the cell type, sequence characteristics of the subtelomeric regions, GC-skew, and alterations in relevant chromatin structure.

Also, as a personal note to the reviewer, we observed that perhaps TERRA expression in BRCA1-deficient cells is also regulated in a manner that is dependent on the telomere maintenance pathway – ALT (U2OS) vs telomerase-positive (HME and HeLa), i.e. there is an opposite trend of TERRA expression on the same chromosomes from ALT vs telomerase-

positive cells. For example, TERRA expression in U2OS cells is high on 7p, 9p, XpYp, and 15q, while it is much lower or nonsignificant on the same chromosomes in HME and HeLa cells. Vice versa, TERRA abundance on 17p and XqYq is higher in HME and HeLa cells compared to U2OS. However, this observation needs further investigation.

For Fig 4:

1. 53BP1 foci is not the best marker for replication stress. Please replace it with phospho-(S317 or S345)-Chk1

This is a great suggestion, and we agree that phosphorylation of Chk1 is a more accurate marker for replication stress. We performed co-staining of TERRA with pChk1^{S345} and TRF2, which can be found in **Figure 5a**.

2. For Figs 4c and 4d, the authors need to show that by overexpression RNaseH1, the pATR foci are reduced, which would then indicate that it is the R-loops that cause the replication stress

As suggested by the reviewer, we performed co-staining of TERRA with pChk1^{S345}/pATR^{Thr1989} and TRF2 in mock-, and RNase H1-treated control and BRCA1-depleted U2OS cells, which can be found in **Figure 5a-c** (respective immunoblot confirming expression of RNase H1-encoding vector is shown in **Supplementary Fig. 6a**). Indeed, an amount of TERRA-pChk1^{S345}/pATR^{Thr1989} co-localized foci fell upon exposure to RNase H1 in BRCA1-depleted cells, suggesting that increased abundance of TERRA R-loops is likely to be the cause of replication stress at telomeres.

For Fig 5:

1. The authors need to demonstrate whether the NLS1 of BRCA1 is important to suppress the replication stress at telomere

To address this question, we performed pChk1^{S345}-TRF2 and pATR^{Thr1989}-TRF2 co-staining experiments in U2OS cells, depleted of endogenous BRCA1 and further transfected with HA-tagged WT and mutant BRCA1 - ΔNLS1 and R506S. We observed increased signs of replication stress at telomeres in cells expressing mutant BRCA1, similar to that observed in BRCA1-depleted cells (**Figure 5d-e, Supplementary Fig. 6g-i**). These results suggest that intact BRCA1 region, responsible for interaction with TERRA, is important for suppression of replication stress at telomeres.

References

1. Kim, J.A., Kruhlak, M., Dotiwala, F., Nussenzweig, A. & Haber, J.E. Heterochromatin is refractory to gamma-H2AX modification in yeast and mammals. *J Cell Biol* **178**, 209-18 (2007).
2. Nakamura, A.J. et al. Both telomeric and non-telomeric DNA damage are determinants of mammalian cellular senescence. *Epigenetics Chromatin* **1**, 6 (2008).
3. Kruhlak, M.J. et al. Changes in chromatin structure and mobility in living cells at sites of DNA double-strand breaks. *J Cell Biol* **172**, 823-34 (2006).

REVIEWERS' COMMENTS

Reviewer #1 (Remarks to the Author):

The authors have done substantial work to answer my questions and have performed all the experiments suggested. With these new data, the manuscript is well improved and my concerns have been addressed. I therefore support this manuscript for publication to Nature Communications.

Reviewer #2 (Remarks to the Author):

The authors carefully addressed all my concerns and performed many additional experiments that, together, contributed to greatly improve the manuscript and to decipher the mechanisms through which BRCA1 regulates TERRA transcription. Over-statements have also been removed. I feel that the manuscript now deserves being published in Nature Communications.

Reviewer #3 (Remarks to the Author):

The authors have addressed all my concerns and questions. Therefore, I recommend its acceptance for publication.

REVIEWERS' COMMENTS

Reviewer #1 (Remarks to the Author):

The authors have done substantial work to answer my questions and have performed all the experiments suggested. With these new data, the manuscript is well improved and my concerns have been addressed. I therefore support this manuscript for publication to Nature Communications.

We would like to thank the reviewer for a positive feedback.

Reviewer #2 (Remarks to the Author):

The authors carefully addressed all my concerns and performed many additional experiments that, together, contributed to greatly improve the manuscript and to decipher the mechanisms through which BRCA1 regulates TERRA transcription. Over-statements have also been removed. I feel that the manuscript now deserves being published in Nature Communications.

We would like to thank the reviewer for a positive feedback.

Reviewer #3 (Remarks to the Author):

The authors have addressed all my concerns and questions. Therefore, I recommend its acceptance for publication.

We would like to thank the reviewer for a positive feedback.